# Learning latent variable structured prediction models with Gaussian perturbations

**Kevin Bello**
Department of Computer Science
Purdue University
West Lafayette, IN, USA
kbellome@purdue.edu

**Jean Honorio**
Department of Computer Science
Purdue University
West Lafayette, IN, USA
jhonorio@purdue.edu

## Abstract

The standard margin-based structured prediction commonly uses a maximum loss over *all* possible structured outputs [26, 1, 5, 25]. The large-margin formulation including latent variables [30, 21] not only results in a non-convex formulation but also increases the search space by a factor of the size of the latent space. Recent work [11] has proposed the use of the maximum loss over *random* structured outputs sampled independently from some proposal distribution, with theoretical guarantees. We extend this work by including latent variables. We study a new family of loss functions under Gaussian perturbations and analyze the effect of the latent space on the generalization bounds. We show that the non-convexity of learning with latent variables originates naturally, as it relates to a tight upper bound of the Gibbs decoder distortion with respect to the latent space. Finally, we provide a formulation using random samples and relaxations that produces a tighter upper bound of the Gibbs decoder distortion up to a statistical accuracy, which enables a polynomial time evaluation of the objective function. We illustrate the method with synthetic experiments and a computer vision application.

## 1 Introduction

Structured prediction is of high interest in many domains such as computer vision [19], natural language processing [32, 33], and computational biology [14]. Some standard methods for structured prediction are conditional random fields (CRFs) [13] and structured SVMs (SSVMs) [25, 26].

In many tasks it is crucial to take into account latent variables. For example, in machine translation, one is usually given a sentence $x$ and its translation $y$, but not the linguistic structure $h$ that connects them (e.g. alignments between words). Even if $h$ is not observable it is important to include this information in the model in order to obtain better prediction results. Examples also arise in computer vision, for instance, most images in indoor scene understanding [28] are cluttered by furniture and decorations, whose appearances vary drastically across scenes, and can hardly be modeled (or even hand-labeled) consistently. In this application, the input $x$ is an image, the structured output $y$ is the layout of the faces (floor, ceiling, walls) and furniture, while the latent structure $h$ assigns a binary label to each pixel (clutter or non-clutter.)

During past years, there has been several solutions to address the problem of latent variables in structured prediction. In the field of computer vision, hidden conditional random fields (HCRF) [23, 29, 22] have been widely applied for object recognition and gesture detection. In natural language processing there is also work in applying discriminative probabilistic latent variable models, for example the training of probabilistic context free grammars with latent annotations in a discriminative manner [20]. The work of Yu and Joachims [30] extends the margin re-scaling SSVM in [26] by introducing latent variables (LSSVM) and obtains a formulation that is optimized using the concave-

convex procedure (CCCP) [31]. The work of Ping et al. [21] considers a smooth objective in LSSVM by incorporating marginal maximum *a posteriori* inference that "averages" over the latent space.

Some of the few works in deriving generalization bounds for structured prediction include the work of McAllester [16], which provides PAC-Bayesian guarantees for arbitrary losses, and the work of Cortes et al. [7], which provides data-dependent margin guarantees for a general family of hypotheses, with an arbitrary factor graph decomposition. However, with the exception of [11], both aforementioned works do not focus on producing computationally appealing methods. Moreover, prior generalization bounds have not focused on latent variables.

**Contributions.**    We focus on the learning aspects of structured prediction problems using latent variables. We first extend the work of [16] by including latent variables, and show that the non-convex formulation using the slack re-scaling approach with latent variables is related to a tight upper bound of the *Gibbs decoder distortion*. This motivates the apparent need of the non-convexity in different formulations using latent variables (e.g., [30, 10]). Second, we provide a tighter upper bound of the Gibbs decoder distortion by randomizing the search space of the optimization problem. That is, instead of having a formulation over all possible structures and latent variables (usually exponential in size), we propose a formulation that uses i.i.d. samples coming from some proposal distribution. This approach is also computationally appealing in cases where the margin can be computed in poly-time (for example, when the latent space is polynomial in size or when a relaxation in the maximization over the latent space can be computed in poly-time), since it would lead to a fully polynomial time evaluation of the formulation. The use of standard Rademacher arguments and the analysis of [11] would lead to a prohibitive upper bound that is proportional to the size of the latent space. We provide a way to obtain an upper bound that is logarithmic in the size of the latent space. Finally, we provide experimental results in synthetic data and in a computer vision application, where we obtain improvements in the average test error with respect to the values reported in [9].

## 2   Background

We denote the input space as $\mathcal{X}$, the output space as $\mathcal{Y}$, and the latent space as $\mathcal{H}$. We assume a distribution $D$ over the observable space $\mathcal{X} \times \mathcal{Y}$. We further assume that we are given a training set $S$ of $n$ i.i.d. samples drawn from the distribution $D$, i.e., $S \sim D^n$.

Let $\mathcal{Y}_x \neq \emptyset$ denote the countable set of feasible outputs or *decodings* of $x$. In general, $|\mathcal{Y}_x|$ is exponential with respect to the input size. Likewise, let $\mathcal{H}_x \neq \emptyset$ denote the countable set of feasible latent decodings of $x$.

We consider a fixed mapping $\Phi$ from triples to feature vectors to describe the relation among input $x$, output $y$, and latent variable $h$, i.e., for any triple $(x, y, h)$, we have the feature vector $\Phi(x, y, h) \in \mathbb{R}^k \setminus \{0\}$. For a parameter $\boldsymbol{w} \in \mathcal{W} \subseteq \mathbb{R}^k \setminus \{0\}$, we consider linear decoders of the form:

$$f_{\boldsymbol{w}}(x) = \operatorname*{argmax}_{(y,h) \in \mathcal{Y}_x \times \mathcal{H}_x} \Phi(x, y, h) \cdot \boldsymbol{w}. \tag{1}$$

The problem of computing this $\mathrm{argmax}$ is typically referred as the *inference* or *prediction* problem. In practice, very few cases of the above general inference problem are tractable, while most are NP-hard and also hard to approximate within a fixed factor. (For instance, see Section 6.1 in [11] for a thorough discussion.)

We denote by $d : \mathcal{Y} \times \mathcal{Y} \times \mathcal{H} \to [0, 1]$ the *distortion* function, which measures the dissimilarity among two elements of the output space $\mathcal{Y}$ and one element of the latent space $\mathcal{H}$. (Note that the distortion function is general in the sense that the latent element may not be used in some applications.) Therefore, the goal is to find a $\boldsymbol{w} \in \mathcal{W}$ that minimizes the decoder distortion, that is:

$$\min_{\boldsymbol{w} \in \mathcal{W}} \mathbb{E}_{(x,y) \sim D} \left[ d(y, \langle f_{\boldsymbol{w}}(x) \rangle) \right]. \tag{2}$$

In the above equation, the angle brackets indicate that we are inserting a pair $(\hat{y}, \hat{h}) = f_{\boldsymbol{w}}(x)$ into the distortion function. From the computational point of view, the above optimization problem is intractable since $d(y, \langle f_{\boldsymbol{w}}(x) \rangle)$ is discontinuous with respect to $\boldsymbol{w}$. From the statistical viewpoint, eq.(2) requires access to the data distribution $D$ and would require an infinite amount of data. In practice, one only has access to a finite number of samples.

Furthermore, even if one were able to compute $\boldsymbol{w}$ using the objective in eq.(2), this parameter $\boldsymbol{w}$, while achieving low distortion, could potentially be in a neighborhood of parameters with high distortion. Therefore, we can optimize a more *robust* objective that takes into account perturbations. In this paper we consider Gaussian perturbations. More formally, let $\alpha > 0$ and let $Q(\boldsymbol{w})$ be a unit-variance Gaussian distribution centered at $\alpha \boldsymbol{w}$ of parameters $\boldsymbol{w}' \in \mathcal{W}$. The Gibbs decoder distortion of the perturbation distribution $Q(\boldsymbol{w})$ and data distribution $D$, is defined as:

$$L(Q(\boldsymbol{w}), D) = \underset{(x,y) \sim D}{\mathbb{E}} \left[ \underset{\boldsymbol{w}' \sim Q(\boldsymbol{w})}{\mathbb{E}} \left[ d(y, \langle f_{\boldsymbol{w}'}(x) \rangle) \right] \right] \tag{3}$$

Then, the optimization problem using the Gibbs decoder distortion can be written as:

$$\min_{\boldsymbol{w} \in W} L(Q(\boldsymbol{w}), D).$$

We define the margin $m(x, y, y', h', \boldsymbol{w})$ as follows:

$$m(x, y, y', h', \boldsymbol{w}) = \max_{h \in \mathcal{H}_x} \Phi(x, y, h) \cdot \boldsymbol{w} - \Phi(x, y', h') \cdot \boldsymbol{w}.$$

Note that since we are considering latent variables, our definition of margin differs from [16, 11]. Let $h^* = \mathrm{argmax}_{h \in \mathcal{H}_x} \Phi(x, y, h) \cdot \boldsymbol{w}$. In this case $h^*$ can be interpreted as the latent variable that best explains the pair $(x, y)$. Then, for a fixed $\boldsymbol{w}$, the margin computes the amount by which the pair $(y, h^*)$ is preferred to the pair $(y', h')$.

Next we introduce the concept of "parts", also used in [16]. Let $c(p, x, y, h)$ be a nonnegative integer that gives the number of times that the part $p \in \mathcal{P}$ appears in the triple $(x, y, h)$. For a part $p \in \mathcal{P}$, we define the feature $p$ as follows:

$$\Phi_p(x, y, h) \equiv c(p, x, y, h)$$

We let $\mathcal{P}_x \neq \emptyset$ denote the set of $p \in \mathcal{P}$ such that there exists $(y, h) \in \mathcal{Y}_x \times \mathcal{H}_x$ with $c(p, x, y, h) > 0$.

**Structural SVMs with latent variables.** [30] extend the formulation of *margin re-scaling* given in [26] incorporating latent variables. The motivation to extend such formulation is that it leads to a difference of two convex functions, which allows the use of CCCP [31]. The aforementioned formulation is:

$$\min_{\boldsymbol{w}} \frac{1}{2} \|\boldsymbol{w}\|_2^2 + C \sum_{(x,y) \in S} \max_{(\hat{y}, \hat{h}) \in \mathcal{Y}_x \times \mathcal{H}_x} [\Phi(x, \hat{y}, \hat{h}) \cdot \boldsymbol{w} + d(y, \hat{y}, \hat{h})] - C \sum_{(x,y) \in S} \max_{h \in \mathcal{H}_x} \Phi(x, y, h) \cdot \boldsymbol{w} \tag{4}$$

In the case of standard SSVMs (without latent variables), [26] discuss two advantages of the *slack re-scaling* formulation over the margin re-scaling formulation, these are: the slack re-scaling formulation is invariant to the scaling of the distortion function, and the margin re-scaling potentially gives significant score to structures that are not even close to being confusable with the target structures. [1, 6, 25] proposed similar formulations to the slack re-scaling formulation. Despite its theoretical advantages, the slack re-scaling has been less popular than the margin re-scaling approach due to computational requirements. In particular, both formulations require optimizing over the output space, but while margin re-scaling preserves the structure of the score and error functions, the slack re-scaling does not. This results in harder inference problems during training. [11] also analyze the slack re-scaling approach and theoretically show that using random structures one can obtain a tighter upper bound of the Gibbs decoder distortion. However, these works do not take into account latent variables.

The following formulation corresponds to the slack re-scaling approach with latent variables:

$$\min_{\boldsymbol{w}} \frac{1}{n} \sum_{(x,y) \in S} \max_{(\hat{y}, \hat{h}) \in \mathcal{Y}_x \times \mathcal{H}_x} d(y, \hat{y}, \hat{h}) \, \mathbb{1}\Big[ m(x, y, \hat{y}, \hat{h}, \boldsymbol{w}) \leq 1 \Big] + \lambda \|\boldsymbol{w}\|_2^2 \tag{5}$$

We take into account the loss of structures whose margin is less than one (i.e., $m(\cdot) \leq 1$) instead of the Hamming distance as done in [11]. This is because the former gave better results in preliminary experiments. Also, it is more related to current practice (e.g., [30]). In order to obtain an SSVM-like formulation, the hinge loss is used instead of the discontinuous $0/1$ loss in the above formulation. Note however, that both eq.(4) and eq.(5) are now non-convex problems with respect to the learning parameter $\boldsymbol{w}$ even if the hinge loss is used.

# 3 The maximum loss over all structured outputs and latent variables

In this section we extend the work of McAllester [16] by including latent variables. In the following theorem, we show that the slack re-scaling objective function (eq.(5)) is an upper bound of the Gibbs decoder distortion (eq.(3)) up to an statistical accuracy of $\mathcal{O}(\sqrt{\log n/n})$ for $n$ training samples.

**Theorem 1.** *Assume that there exists a finite integer value $r$ such that $|\mathcal{Y}_x \times \mathcal{H}_x| \leq r$ for all $(x, y) \in S$. Assume also that $\|\Phi(x, y, h)\|_2 \leq \gamma$ for any triple $(x, y, h)$. Fix $\delta \in (0, 1)$. With probability at least $1 - \delta/2$ over the choice of $n$ training samples, simultaneously for all parameters $\boldsymbol{w} \in \mathcal{W}$ and unit-variance Gaussian perturbation distributions $Q(\boldsymbol{w})$ centered at $\boldsymbol{w}\gamma\sqrt{8 \log (rn/\|\boldsymbol{w}\|_2^2)}$, we have:*

$$L(Q(\boldsymbol{w}), D) \leq \frac{1}{n} \sum_{(x,y) \in S} \max_{(\hat{y},\hat{h}) \in \mathcal{Y}_x \times \mathcal{H}_x} d(y, \hat{y}, \hat{h}) \, \mathbb{1}\Big[m(x, y, \hat{y}, \hat{h}, \boldsymbol{w}) \leq 1\Big] + \frac{\|\boldsymbol{w}\|_2^2}{n}$$

$$+ \sqrt{\frac{4\|\boldsymbol{w}\|_2^2 \, \gamma^2 \log (rn/\|\boldsymbol{w}\|_2^2) + \log (2n/\delta)}{2(n-1)}}$$

(See Appendix A for detailed proofs.)

For the proof of the above we used the PAC-Bayes theorem and well-known Gaussian concentration inequalities. Note that the average sum in the right-hand side, i.e., the objective function, can be equivalently written as:

$$\frac{1}{n} \sum_{(x,y) \in S} \max_{(\hat{y},\hat{h}) \in \mathcal{Y}_x \times \mathcal{H}_x} \min_{h \in \mathcal{H}_x} d(y, \hat{y}, \hat{h}) \, \mathbb{1}\Big[\Phi(x, y, h) \cdot \boldsymbol{w} - \Phi(x, \hat{y}, \hat{h}) \cdot \boldsymbol{w} \leq 1\Big].$$

**Remark 1.** *It is clear that the above formulation is tight with respect to the latent space $\mathcal{H}_x$ due to the minimization. This is an interesting observation because it reinforces the idea that a non-convex formulation is required in models using latent variables, i.e., an attempt to "convexify" the formulation will result in looser upper bounds and consequently might produce worse predictions. Some other examples of non-convex formulations for latent-variable models are found in [30, 10].*

Note also that the upper bound has a maximization over $\mathcal{Y}_x \times \mathcal{H}_x$ (usually exponential in size) and a minimization over $\mathcal{H}_x$ (potentially in exponential size). We state two important observations in the following remark.

**Remark 2.** *First, in the minimization, it is clear that the use of a subset of $\mathcal{H}_x$ would lead to a looser upper bound. However, using a superset $\widetilde{\mathcal{H}}_x \supseteq \mathcal{H}_x$ would lead to a tighter upper bound. The latter relaxation not only can tighten the bound but also can allow the margin to be computed in polynomial time. See for instance some analyses of LP-relaxations in [12, 15, 17]. Second, in contrast, using a subset of $\mathcal{Y}_x \times \mathcal{H}_x$ in the maximization would lead to a tighter upper bound.*

From the first observation above, we will now introduce a new definition of margin, $\widetilde{m}$, which performs a maximization over a superset $\widetilde{\mathcal{H}}_x \supseteq \mathcal{H}_x$.

$$\widetilde{m}(x, y, y', h', \boldsymbol{w}) = \max_{h \in \widetilde{\mathcal{H}}_x} \Phi(x, y, h) \cdot \boldsymbol{w} - \Phi(x, y', h') \cdot \boldsymbol{w}.$$

Several examples are NP-hard $m$ for $\mathcal{H}$ (DAGs, trees or cardinality constrained sets), but poly-time $\widetilde{m}$ for $\widetilde{\mathcal{H}}$ being a set of binary strings. That is, we can encode any DAG (in $\mathcal{H}$) as a binary string (in $\widetilde{\mathcal{H}}$), but not all binary strings are DAGs. Later, in Section 6, we provide an empirical comparison of the use of $m$ and $\widetilde{m}$. We next present a similar upper bound to the one obtained in Theorem 1 but now using the margin $\widetilde{m}$.

**Theorem 2** (Relaxed margin bound.)**.** *Assume that there exists a finite integer value $r$ such that $|\mathcal{Y}_x \times \mathcal{H}_x| \leq r$ for all $(x, y) \in S$. Assume also that $\|\Phi(x, y, h)\|_2 \leq \gamma$ for any triple $(x, y, h)$. Fix $\delta \in (0, 1)$. With probability at least $1 - \delta/2$ over the choice of $n$ training samples, simultaneously for all parameters $\boldsymbol{w} \in \mathcal{W}$ and unit-variance Gaussian perturbation distributions $Q(\boldsymbol{w})$ centered at $\boldsymbol{w}\gamma\sqrt{8 \log (rn/\|\boldsymbol{w}\|_2^2)}$, we have:*

$$L(Q(\boldsymbol{w}), D) \leq \frac{1}{n} \sum_{(x,y) \in S} \max_{(\hat{y},\hat{h}) \in \mathcal{Y}_x \times \mathcal{H}_x} d(y, \hat{y}, \hat{h}) \, \mathbb{1}\Big[\widetilde{m}(x, y, \hat{y}, \hat{h}, \boldsymbol{w}) \leq 1\Big] + \frac{\|\boldsymbol{w}\|_2^2}{n}$$

$$+ \sqrt{\frac{4\|\boldsymbol{w}\|_2^2 \gamma^2 \log\left(rn/\|\boldsymbol{w}\|_2^2\right) + \log\left(2n/\delta\right)}{2(n-1)}}$$

From the second observation in Remark 2, it is natural to ask what elements should constitute this subset in order to control the statistical accuracy with respect to the Gibbs decoder. Finally, if the number of elements is polynomial then we also have an efficient computation of the maximum. We provide answers to these questions in the next section.

## 4 The maximum loss over random structured outputs and latent variables

In this section, we show the relation between PAC-Bayes bounds and the maximum loss over random structured outputs and latent variables sampled i.i.d. from some proposal distribution.

**A more efficient evaluation.** Instead of using a maximization over $\mathcal{Y}_x \times \mathcal{H}_x$, we will perform a maximization over a set $T(\boldsymbol{w}, x)$ of random elements sampled i.i.d. from some proposal distribution $R(\boldsymbol{w}, x)$ with support on $\mathcal{Y}_x \times \mathcal{H}_x$. More explicitly, our new formulation is:

$$\min_{\boldsymbol{w}} \frac{1}{n} \sum_{(x,y)\in S} \max_{(\hat{y},\hat{h})\in T(\boldsymbol{w},x)} d(y,\hat{y},\hat{h}) \; \mathbb{1}\Big[\widetilde{m}(x,y,\hat{y},\hat{h},\boldsymbol{w}) \leq 1\Big] + \lambda\|\boldsymbol{w}\|_2^2. \tag{6}$$

We make use of the following two assumptions in order for $|T(\boldsymbol{w}, x)|$ to be polynomial, even when $|\mathcal{Y}_x \times \mathcal{H}_x|$ is exponential with respect to the input size.

**Assumption A** (Maximal distortion, [11]). *The proposal distribution $R(\boldsymbol{w}, x)$ fulfills the following condition. There exists a value $\beta \in [0, 1)$ such that for all $(x, y) \in S$ and $\boldsymbol{w} \in \mathcal{W}$:*

$$\mathbb{P}_{(y',h')\sim R(\boldsymbol{w},x)}[d(y,y',h') = 1] \geq 1 - \beta$$

**Assumption B** (Low norm). *The proposal distribution $R(\boldsymbol{w}, x)$ fulfills the condition for all $(x, y) \in S$ and $\boldsymbol{w} \in \mathcal{W}$:[1]*

$$\left\| \mathbb{E}_{(y',h')\sim R(\boldsymbol{w},x)} \big[\Phi(x, y, h^*) - \Phi(x, y', h')\big] \right\|_2 \leq \frac{1}{2\sqrt{n}} \leq \frac{1}{2\|\boldsymbol{w}\|_2},$$

*where $h^* = \mathrm{argmax}_{h\in\mathcal{H}_x} \Phi(x, y, h) \cdot \boldsymbol{w}$.*

In Section 5 we provide examples for Assumptions A and B which allow us to obtain $|T(\boldsymbol{w}, x)| = \mathcal{O}\Big(\frac{1}{\log(1/(\beta+e^{-1/(\gamma^2\|\boldsymbol{w}\|_2^2)}))}\Big)$. Note that $\beta$ plays an important role in the number of samples that we need to draw from the proposal distribution $R(\boldsymbol{w}, x)$.

**Statistical analysis.** In this approach, randomness comes from two sources, from the training data $S$ and the random set $T(\boldsymbol{w}, x)$. That is, in Theorem 1, randomness only stems from the training set $S$. Now we need to produce generalization results that hold for all the sets $T(\boldsymbol{w}, x)$, and for all possible proposal distributions $R(\boldsymbol{w}, x)$. The following assumption will allow us to upper-bound the number of possible proposal distributions $R(\boldsymbol{w}, x)$.

**Assumption C** (Linearly inducible ordering, [11]). *The proposal distribution $R(\boldsymbol{w}, x)$ depends solely on the linear ordering induced by the parameter $\boldsymbol{w} \in \mathcal{W}$ and the mapping $\Phi(x, \cdot, \cdot)$. More formally, let $r(x) \equiv |\mathcal{Y}_x \times \mathcal{H}_x|$ and thus $\mathcal{Y}_x \times \mathcal{H}_x \equiv \{(y_1, h_1)\ldots(y_{r(x)}, h_{r(x)})\}$. Let $\boldsymbol{w}, \boldsymbol{w}' \in \mathcal{W}$ be any two arbitrary parameters. Let $\pi(x) = (\pi_1 \ldots \pi_{r(x)})$ be a permutation of $\{1 \ldots r(x)\}$ such that $\Phi(x, y_{\pi_1}, h_{\pi_1}) \cdot \boldsymbol{w} < \cdots < \Phi(x, y_{\pi_{r(x)}}, h_{\pi_{r(x)}}) \cdot \boldsymbol{w}$. Let $\pi'(x) = (\pi'_1 \ldots \pi'_{r(x)})$ be a permutation of $\{1 \ldots r(x)\}$ such that $\Phi(x, y_{\pi'_1}, h_{\pi'_1}) \cdot \boldsymbol{w}' < \cdots < \Phi(x, y_{\pi'_{r(x)}}, h_{\pi'_{r(x)}}) \cdot \boldsymbol{w}'$. For all $\boldsymbol{w}, \boldsymbol{w}' \in \mathcal{W}$ and $x \in \mathcal{X}$, if $\pi(x) = \pi'(x)$ then $\mathbb{KL}\big(R(\boldsymbol{w}, x)\big\|R(\boldsymbol{w}', x)\big) = 0$. In this case, we say that the proposal distribution fulfills $R(\pi(x), x) \equiv R(\boldsymbol{w}, x)$.*

In Assumption C, geometrically speaking, for a fixed $x$ we first project the feature vectors $\Phi(x,y,h)$ of all $(y,h) \in \mathcal{Y}_x \times \mathcal{H}_x$ onto the lines $\boldsymbol{w}$ and $\boldsymbol{w}'$. Let $\pi(x)$ and $\pi'(x)$ be the resulting ordering of the structured outputs after projecting them onto $\boldsymbol{w}$ and $\boldsymbol{w}'$ respectively. Two proposal distributions $R(\boldsymbol{w}, x)$ and $R(\boldsymbol{w}', x)$ are the same provided that $\pi(x) = \pi'(x)$. That is, the specific values of $\Phi(x,y,h) \cdot \boldsymbol{w}$ and $\Phi(x,y,h) \cdot \boldsymbol{w}'$ are irrelevant, and only their ordering matters.

In Section 5 we show an example that fulfills Assumption C, which corresponds to a generalization of Algorithm 2 proposed in [11] for any structure with computationally efficient local changes.

In the following theorem, we show that our new formulation in eq.(6) is related to an upper bound of the Gibbs decoder distortion up to statistical accuracy of $\mathcal{O}(\log^2 n / \sqrt{n})$ for $n$ training samples.

**Theorem 3.** *Assume that there exist finite integer values $r$, $\tilde{r}$, $\ell$, and $\gamma$ such that $|\mathcal{Y}_x \times \mathcal{H}_x| \leq r$ and $|\widetilde{\mathcal{H}}_x| \leq \tilde{r}$ for all $(x,y) \in S$, $|\cup_{(x,y)\in S} \mathcal{P}_x| \leq \ell$, and $\|\Phi(x,y,h)\|_2 \leq \gamma$ for any triple $(x,y,h)$. Assume that the proposal distribution $R(\boldsymbol{w}, x)$ with support on $\mathcal{Y}_x \times \mathcal{H}_x$ fulfills Assumption A with value $\beta$, as well as Assumptions B and C. Assume that $\|\boldsymbol{w}\|_2^2 \leq \frac{1}{128\gamma^2 \log(1/(1-\beta))}$. Fix $\delta \in (0,1)$ and an integer $\mathfrak{s}$ such that $3 \leq 2\mathfrak{s}+1 \leq \frac{9}{20}\sqrt{\ell(r+1)+1}$. With probability at least $1-\delta$ over the choice of both $n$ training samples and $n$ sets of random structured outputs and latent variables, simultaneously for all parameters $\boldsymbol{w} \in \mathcal{W}$ with $\|\boldsymbol{w}\|_0 \leq \mathfrak{s}$, unit-variance Gaussian perturbation distributions $Q(\boldsymbol{w})$ centered at $\boldsymbol{w}\gamma\sqrt{8\log(rn/\|\boldsymbol{w}\|_2^2)}$, and for sets of random structured outputs $T(\boldsymbol{w},x)$ sampled i.i.d. from the proposal distribution $R(\boldsymbol{w},x)$ for each training sample $(x,y) \in S$, such that $|T(\boldsymbol{w},x)| = \left\lceil \frac{1}{2} \frac{\log n}{\log(1/(\beta+e^{-1/(128\gamma^2\|\boldsymbol{w}\|_2^2)}))} \right\rceil$, we have:*

$$L(Q(\boldsymbol{w}), D) \leq \frac{1}{n} \sum_{(x,y)\in S} \max_{(\hat{y},\hat{h})\in T(\boldsymbol{w},x)} d(y,\hat{y},\hat{h}) \, \mathbb{1}\left[\widetilde{m}(x,y,\hat{y},\hat{h},\boldsymbol{w}) \leq 1\right] + \frac{\|\boldsymbol{w}\|_2^2}{n}$$

$$+ \sqrt{\frac{4\|\boldsymbol{w}\|_2^2\gamma^2\log\frac{rn}{\|\boldsymbol{w}\|_2^2} + \log\frac{2n}{\delta}}{2(n-1)}} + \sqrt{\frac{1}{n}} + 3\sqrt{\frac{\mathfrak{s}(\log\ell + 2\log(nr)) + \log(4/\delta)}{n}}$$

$$+ \frac{1}{\log(1/(\beta+e^{-1/(128\gamma^2\|\boldsymbol{w}\|_2^2)}))} \sqrt{\frac{(2\mathfrak{s}+1)\log(\ell(n\tilde{r}+1)+1)\log^3(n+1)}{n}}$$

The proof of the above is based on Theorem 2 as a starting point. In order to account for the computational aspect of requiring sets $T(\boldsymbol{w}, x)$ of polynomial size, we use Assumptions A and B for bounding a *deterministic* expectation. In order to account for the statistical aspects, we use Assumption C and Rademacher complexity arguments for bounding a *stochastic* quantity for all sets $T(\boldsymbol{w}, x)$ of random structured outputs and latent variables, and all possible proposal distributions $R(\boldsymbol{w}, x)$.

**Remark 3.** *A straightforward application of Rademacher complexity in the analysis of [11] leads to a bound of $\mathcal{O}(|\mathcal{H}_x|/\sqrt{n})$. Technically speaking, a classical Rademacher complexity states that: let $\mathcal{F}$ and $\mathcal{G}$ be two hypothesis classes. Let $\min(\mathcal{F}, \mathcal{G}) = \{\min(f,g)|f \in \mathcal{F}, g \in \mathcal{G}\}$. Then $\mathfrak{R}(\min(\mathcal{F},\mathcal{G})) \leq \mathfrak{R}(\mathcal{F}) + \mathfrak{R}(\mathcal{G})$. If we apply this, then Theorem 3 would contain an $\mathcal{O}(|\mathcal{H}_x|/\sqrt{n})$ term, or equivalently $\mathcal{O}(r/\sqrt{n})$. This would be prohibitive since $r$ is typically exponential size, and one would require a very large number of samples $n$ in order to have a useful bound, i.e., to make $\mathcal{O}(r/\sqrt{n})$ close to zero. In the proof we provide a way to tighten the bound to $\mathcal{O}(\sqrt{\log|\mathcal{H}_x|/n})$.*

## 5 Examples

Here we provide several examples that fulfill the three main assumptions of our theoretical result.

**Examples for Assumption A.** First we argue that we can perform a change of measure between different proposal distributions. This allows us to focus on uniform proposals afterwards.

**Claim i** (Change of measure). *Let $R(\boldsymbol{w}, x)$ and $R'(\boldsymbol{w}, x)$ two proposal distributions, both with support on $\mathcal{Y}_x \times \mathcal{H}_x$. Assume that $R(\boldsymbol{w}, x)$ fulfills Assumption A with value $\beta_1$. Let $r_{\boldsymbol{w},x}(\cdot)$ and*

$r'_{\boldsymbol{w},x}(\cdot)$ be the probability mass functions of $R(\boldsymbol{w},x)$ and $R'(\boldsymbol{w},x)$ respectively. Assume that the total variation distance between $R(\boldsymbol{w},x)$ and $R'(\boldsymbol{w},x)$ fulfills for all $(x,y) \in S$ and $\boldsymbol{w} \in \mathcal{W}$:

$$TV(R(\boldsymbol{w},x)\|R'(\boldsymbol{w},x)) \equiv \frac{1}{2} \sum_{(y,h)} |r_{\boldsymbol{w},x}(y,h) - r'_{\boldsymbol{w},x}(y,h)| \leq \beta_2$$

Then $R'(\boldsymbol{w},x)$ fulfills Assumption A with $\beta = \beta_1 + \beta_2$ provided that $\beta_1 + \beta_2 \in [0,1)$.

Next, we present a new result for permutations and for a distortion that returns the number of different positions. We later use this result for an image matching application in the experiments section.

**Claim ii** (Permutations). *Let $\mathcal{Y}_x$ be the set of all permutations of $v$ elements, such that $v > 1$. Let $y_i$ be the $i$-th element in the permutation $y$. Let $d(y,y',h) = \frac{1}{v} \sum_{i=1}^{v} \mathbb{1}\big[y_i \neq y'_i\big]$. The uniform proposal distribution $R(\boldsymbol{w},x) = R(x)$ with support on $\mathcal{Y}_x \times \mathcal{H}_x$ fulfills Assumption A with $\beta = 2/3$.*

The authors in [11] present several examples of distortion functions of the form $d(y,y')$, for directed spanning trees, directed acyclic graphs and cardinality-constrained sets, and a distortion function that returns the number of different edges/elements; as well as for any type of structured output and binary distortion functions. For our setting we can make use of these examples by defining $d(y,y',h) = d(y,y')$. Note that even if we ignore the latent variable in the distortion function, we still use the latent variables in the feature vectors $\Phi(x,y,h)$ and thus in the calculation of the margin.

**Examples for Assumption B.** The claim below is for a particular instance of a sparse mapping and a uniform proposal distribution.

**Claim iii** (Sparse mapping). *Let $b > 0$ be an arbitrary integer value. For all $(x,y) \in S$ with $h^* = \operatorname{argmax}_{h \in \mathcal{H}_x} \Phi(x,y,h) \cdot \boldsymbol{w}$, let $\Upsilon_x = \cup_{p \in \mathcal{P}_x} \Upsilon_x^p$, where the partition $\Upsilon_x^p$ is defined as follows for all $p \in \mathcal{P}_x$:*

$$\Upsilon_x^p \equiv \{(y',h') \mid |\Phi_p(x,y,h^*) - \Phi_p(x,y',h')| \leq b \text{ and } (\forall q \neq p) \, \Phi_q(x,y,h^*) = \Phi_q(x,y',h')\}$$

*If $n \leq |\mathcal{P}_x|/(4b^2)$ for all $(x,y) \in S$, then the uniform proposal distribution $R(\boldsymbol{w},x) = R(x)$ with support on $\mathcal{Y}_x \times \mathcal{H}_x$ fulfills Assumption B.*

The claim below is for a particular instance of a dense mapping and an *arbitrary* proposal distribution.

**Claim iv** (Dense mapping). *Let $b > 0$ be an arbitrary integer value. Let $|\Phi_p(x,y,h^*) - \Phi_p(x,y',h')| \leq \frac{b}{|\mathcal{P}_x|}$ for all $(x,y) \in S$ with $h^* = \operatorname{argmax}_{h \in \mathcal{H}_x} \Phi(x,y,h) \cdot \boldsymbol{w}$, $(y',h') \in \mathcal{Y}_x \times \mathcal{H}_x$ and $p \in \mathcal{P}_x$. If $n \leq |\mathcal{P}_x|/(4b^2)$ for all $(x,y) \in S$, then any arbitrary proposal distribution $R(\boldsymbol{w},x)$ fulfills Assumption B.*

**Examples for Assumption C.** In the case of modeling without latent variables, [32, 33] presented an algorithm for directed spanning trees in the context of dependency parsing in natural language processing. Later, [11] extended the previous algorithm to any structure with computationally efficient local changes, which includes directed acyclic graphs (traversed in post-order) and cardinality-constrained sets. Next, we generalize Algorithm 2 in [11] by including latent variables.

---

**Algorithm 1** Procedure for sampling a structured output $(y',h') \in \mathcal{Y}_x \times \mathcal{H}_x$ from a greedy local proposal distribution $R(\boldsymbol{w},x)$

---
1: **Input:** parameter $\boldsymbol{w} \in \mathcal{W}$, observed input $x \in \mathcal{X}$
2: Draw uniformly at random a structured output $(\hat{y},\hat{h}) \in \mathcal{Y}_x \times \mathcal{H}_x$
3: **repeat**
4:     Make a local change to $(\hat{y},\hat{h})$ in order to increase $\Phi(x,\hat{y},\hat{h}) \cdot \boldsymbol{w}$
5: **until** no refinement in last iteration
6: **Output:** structured output and latent variable $(y',h') \leftarrow (\hat{y},\hat{h})$

---

The above algorithm has the following property:

**Claim v** (Sampling for any type of structured output and latent variable). *Algorithm 1 fulfills Assumption C.*

Table 1: Average over 30 repetitions, and standard error at 95% confidence level. *All (LSSVM)* indicates the use of exact learning and exact inference. *Rand* and *Rand/All* indicate use of random learning, and random and exact inference respectively. *(S)* indicates the use of superset $\widetilde{\mathcal{H}}$ in the calculation of the margin. Rand/All obtains a similar or sightly better test performance than All in the different study cases. Note that the runtime for learning using the randomized approach is much less than exact learning, while still having a good test performance.

| Problem | Method | Training runtime | Training distortion | Test runtime | Test distortion |
|---|---|---|---|---|---|
| Directed spanning trees | All (LSSVM) | $1000 \pm 15$ | $8.4\% \pm 1.4\%$ | $18.9 \pm 0.1$ | $8.2\% \pm 1.3\%$ |
| | Rand (S) | $\mathbf{44 \pm 1}$ | $22\% \pm 2.2\%$ | $0.92 \pm 0$ | $22\% \pm 1.9\%$ |
| | Rand/All (S) | | | $19 \pm 0.1$ | $8.2\% \pm 1.3\%$ |
| | Rand | $126 \pm 5$ | $23\% \pm 3.0\%$ | $3 \pm 0.4$ | $24\% \pm 3.2\%$ |
| | Rand/All | | | $17 \pm 0.8$ | $8.2\% \pm 1.4\%$ |
| Directed acyclic graphs | All (LSSVM) | $1000 \pm 21$ | $17\% \pm 1.7\%$ | $19 \pm 0.2$ | $21\% \pm 2.4\%$ |
| | Rand (S) | $\mathbf{63 \pm 0}$ | $24\% \pm 1.5\%$ | $1.5 \pm 0$ | $28\% \pm 1.9\%$ |
| | Rand/All (S) | | | $19 \pm 0.2$ | $20\% \pm 1.9\%$ |
| | Rand | $353 \pm 5$ | $21\% \pm 1.1\%$ | $8 \pm 1$ | $25\% \pm 1.4\%$ |
| | Rand/All | | | $15 \pm 0.2$ | $19\% \pm 1.6\%$ |
| Cardinality constrained sets | All (LSSVM) | $1000 \pm 5$ | $6.3\% \pm 1.0\%$ | $19.5 \pm 0.1$ | $6\% \pm 1.2\%$ |
| | Rand (S) | $\mathbf{75 \pm 0}$ | $18\% \pm 1.8\%$ | $1.7 \pm 0$ | $18\% \pm 1.8\%$ |
| | Rand/All (S) | | | $19.5 \pm 0.1$ | $6\% \pm 1.3\%$ |
| | Rand | $182 \pm 3$ | $15\% \pm 3.2\%$ | $3.1 \pm 1$ | $17\% \pm 1.2\%$ |
| | Rand/All | | | $19.4 \pm 0.1$ | $6\% \pm 2.2\%$ |

## 6 Experiments

In this section we illustrate the use of our approach by using the formulation in eq.(6). The goal of the synthetic experiments is to show the improvement in prediction results and runtime of our method. While the goal of the real-world experiment is to show the usability of our method in practice.

**Synthetic experiments.** We present experimental results for directed spanning trees, directed acyclic graphs and cardinality-constrained sets. We performed $30$ repetitions of the following procedure. We generated a ground truth parameter $\boldsymbol{w}^*$ with independent zero-mean and unit-variance Gaussian entries. Then, we generated a training set of $n = 100$ samples. Our mapping $\Phi(x, y, h)$ is as follows. For every pair of possible edges/elements $i$ and $j$, we define $\Phi_{ij}(x, y, h) = \mathbb{1}\big[(h_{ij} \text{ xor } x_{ij}) \text{ and } i \in y \text{ and } j \in y\big]$. In order to generate each training sample $(x, y) \in S$, we generated a random vector $x$ with independent Bernoulli entries, each with equal probability of being $1$ or $0$. The latent space $\mathcal{H}$ is the set of binary strings with two entries being $1$, where these two entries share a common edge or element, i.e., $h_{ij} = h_{ik} = 1, \forall i, j, k$. To the best of our knowledge there is no efficient way to *exactly* compute the maximization in the margin $m$ under this latent space. Thus, we define $\widetilde{\mathcal{H}}$ (relaxed set) as the set of all binary strings with exactly *two* entries being $1$. We then can efficiently compute the margin $\tilde{m}$ by a greedy approach since our feature vector is constructed using linear operators. After generating $x$, we set $(y, h) = f_{\boldsymbol{w}^*}(x)$. That is, we solved eq.(1) in order to produce the structured output $y$, and disregard $h$. (More details of the experiment in Appendix B.3.)

We compared three training methods: the maximum loss over *all* possible structured outputs and latent variables with slack re-scaling as in eq.(5). We also evaluated the maximum loss over *random* structured outputs and latent variables, using the original latent space, as well as, the superset relaxation as in eq.(6). We considered directed spanning trees of $4$ nodes, directed acyclic graphs of $4$ nodes and $2$ parents per node, and sets of $3$ elements chosen from $9$ possible elements. After training, for inference on an independent test set, we used eq.(1) for the maximum loss over *all* possible structured outputs and latent variables. For the maximum loss over random structured outputs and latent variables, we use the following *approximate* inference approach:

$$\widetilde{f}_{\boldsymbol{w}}(x) \equiv \operatorname*{argmax}_{(y,h) \in T(\boldsymbol{w},x)} \Phi(x, y, h) \cdot \boldsymbol{w}. \tag{7}$$

Note that we used small structures and latent spaces in order to compare to exact learning, i.e., going through all possible structures as in eq.(5) and eq.(4). Bigger structures would result in exponential number of structures, making exact methods intractable to compare against our method. For purposes

of testing, we tried cardinality constrained sets of $4$ elements out of $100$ (note that in this case $|\mathcal{Y}| \approx 10^8$, $|\mathcal{H}| \approx 10^{16}$) and training only took $11$ minutes under our approach.

Table 1 shows the runtime, the training distortion as well as the test distortion in an independently generated set of $100$ samples. In the different study cases, the maximum loss over *random* structured outputs and latent variables obtains similar test performance than the maximum loss over *all* possible structured outputs and latent variables. However, note that our method is considerable faster.

**Image matching.** We illustrate our approach for image matching on video frames from the Buffy Stickmen dataset (`http://www.robots.ox.ac.uk/~vgg/data/stickmen/`). The goal of the experiment is to match the keypoints representing different body parts, between two images. Each frame contains $18$ keypoints representing different parts of the body. From a total of $187$ image pairs (from different episodes and people), we randomly selected $120$ pairs for training and the remaining $67$ pairs for testing. We performed $30$ repetitions. Ground truth keypoint matching is provided in the dataset.

Following [9, 27], we represent the matching as a permutation of keypoints. Let $x = (I, I')$ be a pair of images, and let $y$ be a permutation of $\{1 \ldots 18\}$. We model the latent variable $h$ as a $\mathbb{R}^{2 \times 2}$ matrix representing an affine transformation of a keypoint, where $h_{11}, h_{22} \in \{0.8, 1, 1.2\}$, and $h_{12}, h_{21} \in \{-0.2, 0, 0.2\}$. Our mapping $\Phi(x, y, h)$ uses SIFT features, and the distance between coordinates after using $h$. (Details in Appendix B.3.)

We used the distortion function and $\beta = 2/3$ as prescribed by Claim ii. After learning, for a given $x$ from the test set, we performed $100$ iterations of random inference as in eq.(7). We obtained an average error of $0.3878$ ($6.98$ incorrectly matched keypoints) in the test set, which is an improvement to the values of $8.47$ for maximum-a-posteriori perturbations and $8.69$ for max-margin, as reported in [9]. Finally, we show an example from the test set in Figure 1.

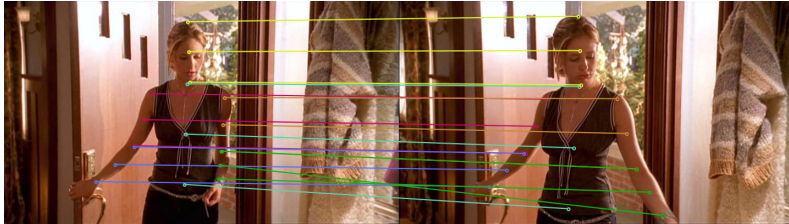

Figure 1: Image matching on the Buffy Stickmen dataset, predicted by our randomized approach with latent variables. The problem is challenging since the dataset contains different episodes and people.

## 7 Future directions

The randomization of the latent space in the calculation of the margin is of high interest. Despite leading to a looser upper bound of the Gibbs decoder distortion, if one could control the statistical accuracy under this approach then one could obtain a fully polynomial-time evaluation of the objective function, even if $|\mathcal{H}|$ is exponential. Therefore, whether this method is feasible, and under what technical conditions, are potential future work. The analysis of other non-Gaussian perturbation models from the computational and statistical viewpoints is also of interest. Finally, it would be interesting to analyze *approximate* inference for prediction on an independent test set.

## Acknowledgments

This material is based upon work supported by the National Science Foundation under Grant No. 1716609-IIS.

## Footnotes

[1]The second inequality follows from an implicit assumption made in Theorem 1, i.e., $\|\boldsymbol{w}\|_2^2 /n \leq 1$ since the distortion function $d$ is at most 1.

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
