[Supplementary Material]

# SUPPLEMENTARY MATERIAL
## Learning latent variable structured prediction models with Gaussian perturbations

## Appendix A  Detailed Proofs

In this section, we state the proofs of all the theorems in our manuscript.

### A.1  Proof of Theorem 1

Here, we provide the proof of Theorem 1. First, we derive an intermediate lemma needed for the final proof.

**Lemma 1** (Adapted from Lemma 5 in [16]). *Assume that there exists a finite integer value $r$ such that, $|\mathcal{Y}_x \times \mathcal{H}_x| \leq r$ for all $(x, y) \in S$. Assume also that $\|\Phi(x, y, h)\|_2 \leq \gamma$ for any triple $(x, y, h)$. Let $Q(\boldsymbol{w})$ be a unit-variance Gaussian distribution centered at $\alpha \boldsymbol{w}$ for $\alpha = \gamma\sqrt{8 \log \frac{rn}{\|\boldsymbol{w}\|_2^2}}$. Then for all $(x, y) \in S$, and all $\boldsymbol{w} \in \mathcal{W}$, we have:*

$$\mathbb{P}_{\boldsymbol{w}' \sim Q(\boldsymbol{w})}[m(x, y, \langle f_{\boldsymbol{w}'}(x) \rangle), \boldsymbol{w}) \geq 1] \leq \|\boldsymbol{w}\|_2^2/n$$

*or equivalently:*

$$\mathbb{P}_{\boldsymbol{w}' \sim Q(\boldsymbol{w})}[m(x, y, \langle f_{\boldsymbol{w}'}(x) \rangle), \boldsymbol{w}) \leq 1] \geq 1 - \|\boldsymbol{w}\|_2^2/n \tag{8}$$

*Proof.* Note that the randomness in the statement comes from the variable $\boldsymbol{w}'$, then by a union bound on the elements of $\mathcal{Y}_x \times \mathcal{H}_x$ it suffices to show that for any given $(\hat{y}, \hat{h})$ with $m(x, y, \hat{y}, \hat{h}, \boldsymbol{w}) \geq 1$, the probability that $f_{\boldsymbol{w}'}(x) = (\hat{y}, \hat{h})$ is at most $\|\boldsymbol{w}\|_2^2/(rn)$.

Consider a fixed $(\hat{y}, \hat{h}) \in \mathcal{Y}_x \times \mathcal{H}_x$ with $m(x, y, \hat{y}, \hat{h}, \boldsymbol{w}) \geq 1$. First, by well-know concentration inequalities we have that for any vector $\Psi \in \mathbb{R}^\ell$ with $\|\Psi\|_2 = 1$ and $\varepsilon \geq 0$:

$$\mathbb{P}_{\boldsymbol{w}' \sim Q(\boldsymbol{w})}[(\alpha \boldsymbol{w} - \boldsymbol{w}') \cdot \Psi \geq \varepsilon] \leq e^{-\varepsilon^2/2} \tag{9}$$

Let $h^* = \operatorname{argmax}_{h \in \mathcal{H}_x} \Phi(x, y, h) \cdot \boldsymbol{w}$, and let $\Delta(x, y, h^*, \hat{y}, \hat{h}) = \Phi(x, y, h^*) - \Phi(x, \hat{y}, \hat{h})$. Then, $m(x, y, \hat{y}, \hat{h}, \boldsymbol{w}) = \Delta(x, y, h^*, \hat{y}, \hat{h}) \cdot \boldsymbol{w}$. Using $\Psi = \Delta(x, y, h^*, \hat{y}, \hat{h})/\|\Delta(x, y, h^*, \hat{y}, \hat{h})\|_2$ in (9) we have:

$$\mathbb{P}_{\boldsymbol{w}' \sim Q(\boldsymbol{w})}[m(x, y, \hat{y}, \hat{h}, \boldsymbol{w}') \leq \alpha m(x, y, \hat{y}, \hat{h}, \boldsymbol{w}) - \varepsilon\|\Delta(x, y, h^*, \hat{y}, \hat{h})\|_2] \leq e^{-\varepsilon^2/2}$$

$$\mathbb{P}_{\boldsymbol{w}' \sim Q(\boldsymbol{w})}[m(x, y, \hat{y}, \hat{h}, \boldsymbol{w}') \leq \alpha - \varepsilon\|\Delta(x, y, h^*, \hat{y}, \hat{h})\|_2] \leq e^{-\varepsilon^2/2}$$

$$\mathbb{P}_{\boldsymbol{w}' \sim Q(\boldsymbol{w})}[m(x, y, \hat{y}, \hat{h}, \boldsymbol{w}') \leq 0] \leq e^{-\alpha^2/(8\gamma^2)} \tag{10.a}$$

$$\mathbb{P}_{\boldsymbol{w}' \sim Q(\boldsymbol{w})}[f_{w'}(x) = (\hat{y}, \hat{h})] \leq e^{-\alpha^2/(8\gamma^2)}$$

where the step in (10.a) follows from $\varepsilon = \alpha/\|\Delta(x, y, h^*, \hat{y}, \hat{h})\|_2$ and $\|\Delta(x, y, h^*, \hat{y}, \hat{h})\|_2 \leq 2\gamma$. Thus, we prove our claim. $\qquad\square$

Next, we provide the final proof.

*Proof of Theorem 1.* Define the Gibbs decoder *empirical* distortion of the perturbation distribution $Q(\boldsymbol{w})$ and training set $S$ as:

$$L(Q(\boldsymbol{w}), S) = \frac{1}{n} \sum_{(x,y) \in S} \mathbb{E}_{\boldsymbol{w}' \sim Q(\boldsymbol{w})} [d(y, \langle f_{\boldsymbol{w}'}(x) \rangle)]$$

In PAC-Bayes terminology, $Q(\boldsymbol{w})$ is the *posterior* distribution. Let the *prior* distribution $P$ be the unit-variance zero-mean Gaussian distribution. Fix $\delta \in (0,1)$ and $\alpha > 0$. By well-known PAC-Bayes proof techniques, Lemma 4 in [16] shows that with probability at least $1 - \delta/2$ over the choice of $n$ training samples, simultaneously for all parameters $\boldsymbol{w} \in \mathcal{W}$, and unit-variance Gaussian posterior distributions $Q(\boldsymbol{w})$ centered at $\boldsymbol{w}\alpha$, we have:

$$L(Q(\boldsymbol{w}), D) \leq L(Q(\boldsymbol{w}), S) + \sqrt{\frac{KL(Q(\boldsymbol{w}) \| P) + \log(2n/\delta)}{2(n-1)}}$$

$$= L(Q(\boldsymbol{w}), S) + \sqrt{\frac{\|w\|_2^2 \, \alpha^2/2 + \log(2n/\delta)}{2(n-1)}} \qquad (11)$$

Thus, an upper bound of $L(Q(\boldsymbol{w}), S)$ would lead to an upper bound of $L(Q(\boldsymbol{w}), D)$. In order to upper-bound $L(Q(\boldsymbol{w}), S)$, we can upper-bound each of its summands, i.e., we can upper-bound $\mathbb{E}_{\boldsymbol{w}' \sim Q(\boldsymbol{w})}[d(y, f_{\boldsymbol{w}'}(x))]$ for each $(x, y) \in S$. Define the distribution $Q(\boldsymbol{w}, x)$ with support on $\mathcal{Y}_x \times \mathcal{H}_x$ in the following form for all $y \in \mathcal{Y}_x$ and $h \in \mathcal{H}_x$:

$$\mathbb{P}_{(y',h') \sim Q(\boldsymbol{w}, x)} [(y', h') = (y, h)] \equiv \mathbb{P}_{\boldsymbol{w}' \sim Q(\boldsymbol{w})} [f_{\boldsymbol{w}'}(x) = (y, h)] \qquad (12)$$

For clarity of presentation, define:

$$u(x, y, y', h', \boldsymbol{w}) \equiv 1 - m(x, y, y', h', \boldsymbol{w})$$

Let $u \equiv u(x, y, \langle f_{\boldsymbol{w}'}(x) \rangle, \boldsymbol{w})$. Simultaneously for all $(x, y) \in S$, we have:

$$\mathbb{E}_{\boldsymbol{w}' \sim Q(\boldsymbol{w})} \big[ d(y, \langle f_{\boldsymbol{w}'}(x) \rangle) \big] = \mathbb{E}_{\boldsymbol{w}' \sim Q(\boldsymbol{w})} \big[ d(y, \langle f_{\boldsymbol{w}'}(x) \rangle) \, \mathbb{1}[u \geq 0] + d(y, \langle f_{\boldsymbol{w}'}(x) \rangle) \, \mathbb{1}[u < 0] \big]$$

$$\leq \mathbb{E}_{\boldsymbol{w}' \sim Q(\boldsymbol{w})} \big[ d(y, \langle f_{\boldsymbol{w}'}(x) \rangle) \, \mathbb{1}[u \geq 0] + \mathbb{1}[u < 0] \big] \qquad (13.a)$$

$$= \mathbb{E}_{\boldsymbol{w}' \sim Q(\boldsymbol{w})} \big[ d(y, \langle f_{\boldsymbol{w}'}(x) \rangle \, \mathbb{1}[u \geq 0] \big] + \mathbb{P}_{\boldsymbol{w}' \sim Q(\boldsymbol{w})} [u < 0]$$

$$\leq \mathbb{E}_{\boldsymbol{w}' \sim Q(\boldsymbol{w})} \big[ d(y, \langle f_{\boldsymbol{w}'}(x) \rangle \, \mathbb{1}[u \geq 0] \big] + \|\boldsymbol{w}\|_2^2 / n \qquad (13.b)$$

$$= \mathbb{E}_{\boldsymbol{w}' \sim Q(\boldsymbol{w})} \big[ d(y, \langle f_{\boldsymbol{w}'}(x) \rangle \, \mathbb{1} \big[ u(x, y, \langle f_{\boldsymbol{w}'}(x) \rangle, \boldsymbol{w}) \geq 0 \big] \big] + \|\boldsymbol{w}\|_2^2 / n$$

$$= \mathbb{E}_{(y',h') \sim Q(\boldsymbol{w}, x)} \big[ d(y, y', h') \, \mathbb{1} \big[ u(x, y, y', h', \boldsymbol{w}) \geq 0 \big] \big] + \|\boldsymbol{w}\|_2^2 / n$$
$$(13.c)$$

$$\leq \max_{(\hat{y}, \hat{h}) \in \mathcal{Y}_x \times \mathcal{H}_x} d(y, \hat{y}, \hat{h}) \, \mathbb{1} \big[ u(x, y, \hat{y}, \hat{h}, \boldsymbol{w}) \geq 0 \big] + \|\boldsymbol{w}\|_2^2 / n \qquad (13.d)$$

where the step in eq.(13.a) holds since $d : \mathcal{Y} \times \mathcal{Y} \times \mathcal{H} \to [0, 1]$. The step in eq.(13.b) follows from Lemma 1 which states that $\mathbb{P}_{\boldsymbol{w}' \sim Q(\boldsymbol{w})}[u(x, y, \langle f_{\boldsymbol{w}'}(x) \rangle, \boldsymbol{w}) < 0] \leq \|\boldsymbol{w}\|_2^2 / n$ for $\alpha = \gamma \sqrt{8 \log(rn/\|\boldsymbol{w}\|_2^2)}$, for all $(x, y) \in S$ and all $\boldsymbol{w} \in \mathcal{W}$. By the definition in eq.(12), then the step in eq.(13.c) holds. Let $\lambda : \mathcal{Y} \times \mathcal{H} \to [0, 1]$ be some arbitrary function, the step in eq.(13.d) uses the fact that $\mathbb{E}_{(y,h)}[\lambda(y, h)] \leq \max_{(y,h)} \lambda(y, h)$.

By eq.(11) and eq.(13.d), we prove our claim. $\qquad \square$

## A.2  Proof of Theorem 2

*Proof.* The proof follows similar steps to that of Theorem 1. Note that the relaxed margin, $\widetilde{m}$, also fulfills the bound in Lemma 1. Hence, following the steps of Proof A.1 we obtain an upper bound with same constants. $\qquad \square$

### A.3 Proof of Theorem 3

Here, we provide the proof of Theorem 3. First, we derive an intermediate lemma needed for the final proof.

**Lemma 2.** *Let $\Delta \in \mathbb{R}^\ell$ be a random variable with $\|\Delta\|_2 \leq 2\gamma$, and $\boldsymbol{w} \in \mathbb{R}^\ell$ be a constant. If $\mathbb{E}[\Delta] \cdot \boldsymbol{w} \leq 1/2$ then we have:*

$$\mathbb{P}[\Delta \cdot \boldsymbol{w} > 1] \leq \exp\left(\frac{-1}{128\gamma^2\|\boldsymbol{w}\|_2^2}\right)$$

*Proof.* Let $t > 0$, we have that:

$$\mathbb{P}[\Delta \cdot \boldsymbol{w} > 1] = \mathbb{P}[(\Delta - \mathbb{E}[\Delta]) \cdot \boldsymbol{w} > 1 - \mathbb{E}[\Delta] \cdot \boldsymbol{w}]$$
$$\leq \mathbb{P}[(\Delta - \mathbb{E}[\Delta]) \cdot \boldsymbol{w} \geq 1/2] \tag{14.b}$$
$$= \mathbb{P}[\exp\left(t(\Delta - \mathbb{E}[\Delta]) \cdot \boldsymbol{w}\right) \geq e^{t/2}]$$
$$\leq e^{-t/2} \, \mathbb{E}[\exp\left(t(\Delta - \mathbb{E}[\Delta]) \cdot \boldsymbol{w}\right)] \tag{14.c}$$
$$\leq \exp\left(-t/2 + 8t^2\gamma^2\|\boldsymbol{w}\|_2^2\right) \tag{14.d}$$

The step in eq.(14.b) follows from $\mathbb{E}[\Delta] \cdot \boldsymbol{w} \leq 1/2$ and thus $1 - \mathbb{E}[\Delta] \cdot \boldsymbol{w} \geq 1/2$. The step in eq.(14.c) follows from Markov's inequality. The step in eq.(14.d) follows from Hoeffding's lemma and the fact that the random variable $z = (\Delta - \mathbb{E}[\Delta]) \cdot \boldsymbol{w}$ fulfills $\mathbb{E}[z] = 0$ as well as $z \in [-4\gamma\|\boldsymbol{w}\|_2, +4\gamma\|\boldsymbol{w}\|_2]$. In more detail, note that $\|\Delta\|_2 \leq 2\gamma$ and by Jensen's inequality $\|\mathbb{E}[\Delta]\|_2 \leq \mathbb{E}[\|\Delta\|_2] \leq 2\gamma$. Then, note that by Cauchy-Schwarz inequality $|(\Delta - \mathbb{E}[\Delta]) \cdot \boldsymbol{w}| \leq \|\Delta - \mathbb{E}[\Delta]\|_2\|\boldsymbol{w}\|_2 \leq (\|\Delta\|_2 + \|\mathbb{E}[\Delta]\|_2)\|\boldsymbol{w}\|_2 \leq 4\gamma\|\boldsymbol{w}\|_2$. Finally, let $g(t) = -t/2 + 8t^2\gamma^2\|\boldsymbol{w}\|_2^2$. By making $\partial g/\partial t = 0$, we get the optimal setting $t^* = 1/(32\gamma^2\|\boldsymbol{w}\|_2^2)$. Thus, $g(t^*) = -1/(128\gamma^2\|\boldsymbol{w}\|_2^2)$ and we prove our claim. □

Next, we provide the final proof.

*Proof of Theorem 3.* Note that sampling from the distribution $Q(\boldsymbol{w}, x)$ as defined in eq.(12) is NP-hard in general, thus our plan is to upper-bound the expectation in eq.(13.c) by using the maximum over random structured outputs and latent variables sampled independently from a proposal distribution $R(\boldsymbol{w}, x)$ with support on $\mathcal{Y}_x \times \mathcal{H}_x$.

Let $T(\boldsymbol{w}, x)$ be a set of $n'$ i.i.d. random structured outputs and latent variables drawn from the proposal distribution $R(\boldsymbol{w}, x)$, i.e., $T(\boldsymbol{w}, x) \sim R(\boldsymbol{w}, x)^{n'}$. Furthermore, let $\mathbb{T}(\boldsymbol{w})$ be the collection of the $n$ sets $T(\boldsymbol{w}, x)$ for all $(x, y) \in S$, i.e. $\mathbb{T}(\boldsymbol{w}) \equiv \{T(\boldsymbol{w}, x)\}_{(x,y)\in S}$ and thus $\mathbb{T}(w) \sim \{R(\boldsymbol{w}, x)^{n'}\}_{(x,y)\in S}$. For clarity of presentation, define:

$$v(x, y, y', h', \boldsymbol{w}) \equiv d(y, y', h') \, \mathbb{1}\big[\widetilde{m}(x, y, y', h', \boldsymbol{w}) \leq 1\big]$$

For sets $T(\boldsymbol{w}, x)$ of sufficient size $n'$, our goal is to upper-bound eq.(13.c) in the following form for all parameters $\boldsymbol{w} \in \mathcal{W}$:

$$\frac{1}{n}\sum_{(x,y)\in S}\mathbb{E}_{(y',h')\sim Q(\boldsymbol{w},x)}[v(x, y, y', h', \boldsymbol{w})] \leq \frac{1}{n}\sum_{(x,y)\in S}\max_{(\hat{y},\hat{h})\in T(\boldsymbol{w},x)}v(x, y, \hat{y}, \hat{h}, \boldsymbol{w}) + \mathcal{O}(\log^2 n/\sqrt{n})$$

Note that the above expression would produce a tighter upper bound than the maximum loss over all possible structured outputs and latent variables since $\max_{(\hat{y},\hat{h})\in T(\boldsymbol{w},x)} v(x, y, \hat{y}, \hat{h}, \boldsymbol{w}) \leq \max_{(\hat{y},\hat{h})\in\mathcal{Y}_x\times\mathcal{H}_x} v(x, y, \hat{y}, \hat{h}, \boldsymbol{w})$. For analysis purposes, we decompose the latter equation into two quantities:

$$A(\boldsymbol{w}, S) \equiv \frac{1}{n}\sum_{(x,y)\in S}\left(\mathbb{E}_{(y',h')\sim Q(\boldsymbol{w},x)}[v(x, y, y', h', \boldsymbol{w})] - \mathbb{E}_{T(\boldsymbol{w},x)\sim R(\boldsymbol{w},x)^{n'}}\left[\max_{(\hat{y},\hat{h})\in T(\boldsymbol{w},x)}v(x, y, \hat{y}, \hat{h}, \boldsymbol{w})\right]\right)$$
$$\tag{15}$$

$$B(\boldsymbol{w}, S, \mathbb{T}(\boldsymbol{w})) \equiv \frac{1}{n} \sum_{(x,y)\in S} \left( \mathop{\mathbb{E}}_{T(\boldsymbol{w},x)\sim R(\boldsymbol{w},x)^{n'}} \left[ \max_{(\hat{y},\hat{h})\in T(\boldsymbol{w},x)} v(x,y,\hat{y},\hat{h},\boldsymbol{w}) \right] - \max_{(\hat{y},\hat{h})\in T(\boldsymbol{w},x)} v(x,y,\hat{y},\hat{h},\boldsymbol{w}) \right)$$
(16)

Thus, we will show that $A(\boldsymbol{w}, S) \leq \sqrt{1/n}$ and $B(\boldsymbol{w}, S, \mathbb{T}(\boldsymbol{w})) \leq \mathcal{O}(\log^2 n/\sqrt{n})$ for all parameters $\boldsymbol{w} \in \mathcal{W}$, any training set $S$ and all collections $\mathbb{T}(\boldsymbol{w})$, and therefore $A(\boldsymbol{w}, S) + B(\boldsymbol{w}, S, \mathbb{T}(\boldsymbol{w})) \leq \mathcal{O}(\log^2 n/\sqrt{n})$. Note that while the value of $A(\boldsymbol{w}, S)$ is deterministic, the value of $B(\boldsymbol{w}, S, \mathbb{T}(\boldsymbol{w}))$ is stochastic given that $\mathbb{T}(\boldsymbol{w})$ is a collection of sampled random structured outputs.

Fix a specific $\boldsymbol{w} \in \mathcal{W}$. If data is separable then $v(x,y,y',h',\boldsymbol{w}) = 0$ for all $(x,y) \in S$ and $(y',h') \in \mathcal{Y}_x \times \mathcal{H}_x$. Thus, we have $A(\boldsymbol{w}, S) = B(\boldsymbol{w}, S, \mathbb{T}(\boldsymbol{w})) = 0$ and we complete our proof for the separable case.[2] In what follows, we focus on the non-separable case.

**Bounding the Deterministic Expectation** $A(\boldsymbol{w}, S)$**.** Here, we show that in eq.(15), $A(\boldsymbol{w}, S) \leq \sqrt{1/n}$ for all parameters $\boldsymbol{w} \in \mathcal{W}$ and any training set $S$, provided that we use a sufficient number $n'$ of random structured outputs sampled from the proposal distribution.

By well-known identities, we can rewrite:

$$A(\boldsymbol{w}, S) = \frac{1}{n} \sum_{(x,y)\in S} \int_0^1 \left( \mathop{\mathbb{P}}_{(y',h')\sim R(\boldsymbol{w},x)}[v(x,y,y',h',\boldsymbol{w}) < z]^{n'} - \mathop{\mathbb{P}}_{(y',h')\sim Q(\boldsymbol{w},x)}[v(x,y,y',h',\boldsymbol{w}) < z] \right) dz$$
(17.a)

$$\leq \frac{1}{n} \sum_{(x,y)\in S} \mathop{\mathbb{P}}_{(y',h')\sim R(\boldsymbol{w},x)}[v(x,y,y',h',\boldsymbol{w}) < 1]^{n'}$$

$$= \frac{1}{n} \sum_{(x,y)\in S} \mathop{\mathbb{P}}_{(y',h')\sim R(\boldsymbol{w},x)}[d(y,y',h') < 1 \vee \widetilde{m}(x,y,y',h',\boldsymbol{w}) > 1]^{n'}$$

$$\leq \frac{1}{n} \sum_{(x,y)\in S} \left( \left( 1 - \mathop{\mathbb{P}}_{(y',h')\sim R(\boldsymbol{w},x)}[d(y,y',h') = 1] \right) + \mathop{\mathbb{P}}_{(y',h')\sim R(\boldsymbol{w},x)}[\widetilde{m}(x,y,y',h',\boldsymbol{w}) > 1] \right)^{n'}$$

$$\leq \left( \beta + \exp\left( \frac{-1}{128\gamma^2\|\boldsymbol{w}\|_2^2} \right) \right)^{n'}$$
(17.b)

$$= \sqrt{1/n}$$
(17.c)

where the step in eq.(17.a) holds since for two independent random variables $g, h \in [0,1]$, we have $\mathbb{E}[g] = 1 - \int_0^1 \mathbb{P}[g < z] dz$ and $\mathbb{P}[\max(g,h) < z] = \mathbb{P}[g < z]\,\mathbb{P}[h < z]$. Therefore, $\mathbb{E}[\max(g,h)] = 1 - \int_0^1 \mathbb{P}[g < z]\,\mathbb{P}[h < z] dz$. For the step in eq.(17.b), we used Assumption A for the first term in the sum. For the second term in the sum, let $\Delta \equiv \Phi(x,y,h^*) - \Phi(x,y',h')$ where $h^* = \operatorname{argmax}_{h\in\widetilde{\mathcal{H}}_x} \Phi(x,y,h) \cdot \boldsymbol{w}$, then $\widetilde{m}(x,y,y',h',\boldsymbol{w}) = \Delta \cdot \boldsymbol{w}$. From $\|\Phi(x,y,h)\|_2 \leq \gamma$, we have that $\|\Delta\|_2 \leq 2\gamma$. By Assumption B, we have that $\|\mathbb{E}[\Delta]\|_2 \leq 1/(2\sqrt{n}) \leq 1/(2\|\boldsymbol{w}\|_2)$. By Cauchy-Schwarz inequality we have $\mathbb{E}[\Delta] \cdot \boldsymbol{w} \leq \|\mathbb{E}[\Delta]\|_2 \|\boldsymbol{w}\|_2 \leq \|\boldsymbol{w}\|_2 / (2\|\boldsymbol{w}\|_2) \leq 1/2$. Since $\mathbb{E}[\Delta] \cdot \boldsymbol{w} \leq 1/2$ and $\|\Delta\|_2 \leq 2\gamma$, we apply Lemma 2 in the step in eq.(17.b). For the step in eq.(17.c), let $\lambda \equiv \frac{1}{\log(1/(\beta+e^{-1/(128\gamma^2\|\boldsymbol{w}\|_2^2)}))}$. Furthermore, let $n' = \frac{1}{2}\lambda \log n$. Therefore,

$$\left( \beta + \exp\left( \frac{-1}{128\gamma^2\|\boldsymbol{w}\|_2^2} \right) \right)^{n'} = \sqrt{1/n}.$$

**Bounding the Stochastic Quantity** $B(\boldsymbol{w}, S, \mathbb{T}(\boldsymbol{w}))$**.** Here, we show that in eq.(16), $B(\boldsymbol{w}, S, \mathbb{T}(\boldsymbol{w})) \leq \mathcal{O}(\log^2 n/\sqrt{n})$ for all parameters $\boldsymbol{w} \in \mathcal{W}$, any training set $S$ and all collections

$\mathbb{T}(\boldsymbol{w})$. For clarity of presentation, define:

$$g(x, y, T, \boldsymbol{w}) \equiv \max_{(\hat{y}, \hat{h}) \in T} v(x, y, \hat{y}, \hat{h}, \boldsymbol{w})$$

Thus, we can rewrite:

$$B(\boldsymbol{w}, S, \mathbb{T}(\boldsymbol{w})) = \frac{1}{n} \sum_{(x,y) \in S} \left( \mathop{\mathbb{E}}_{T(\boldsymbol{w},x) \sim R(\boldsymbol{w},x)^{n'}} [g(x, y, T(\boldsymbol{w}, x), \boldsymbol{w})] - g(x, y, T(\boldsymbol{w}, x), \boldsymbol{w}) \right)$$

Let $r_x \equiv |\mathcal{Y}_x \times \mathcal{H}_x|$ and thus $\mathcal{Y}_x \times \mathcal{H}_x \equiv \{(y_1, h_1) \ldots (y_{r_x}, h_{r_x})\}$. Let $\pi(x) = (\pi_1 \ldots \pi_{r_x})$ be a permutation of $\{1 \ldots r_x\}$ such that $\Phi(x, y_{\pi_1}, h_{\pi_1}) \cdot \boldsymbol{w} < \cdots < \Phi(x, y_{\pi_{r_x}}, h_{\pi_{r_x}}) \cdot \boldsymbol{w}$. Let $\Pi$ be the collection of the $n$ permutations $\pi(x)$ for all $(x, y) \in S$, i.e. $\Pi = \{\pi(x)\}_{(x,y) \in S}$. From Assumption C, we have that $R(\pi(x), x) \equiv R(\boldsymbol{w}, x)$. Similarly, we rewrite $T(\pi(x), x) \equiv T(\boldsymbol{w}, x)$ and $\mathbb{T}(\Pi) \equiv \mathbb{T}(\boldsymbol{w})$.

Furthermore, let $\mathcal{W}_{\Pi,S}$ be the set of all $\boldsymbol{w} \in \mathcal{W}$ that induce $\Pi$ on the training set $S$. For the parameter space $\mathcal{W}$, collection $\Pi$ and training set $S$, define the function class $\mathfrak{G}_{\mathcal{W},\Pi,S}$ as follows:

$$\mathfrak{G}_{\mathcal{W},\Pi,S} \equiv \{g(x, y, T, \boldsymbol{w}) \mid \boldsymbol{w} \in \mathcal{W}_{\Pi,S} \text{ and } (x, y) \in S\}$$

Note that since $|\mathcal{Y}_x \times \mathcal{H}_x| \leq r$ for all $(x, y) \in S$, then $|\cup_{(x,y) \in S} \mathcal{Y}_x \times \mathcal{H}_x| \leq \sum_{(x,y) \in S} |\mathcal{Y}_x \times \mathcal{H}_x| \leq nr$. Note that each ordering of the $nr$ structured outputs completely determines a collection $\Pi$ and thus the collection of proposal distributions $R(\boldsymbol{w}, x)$ for each $(x, y) \in S$. Note that since $|\cup_{(x,y) \in S} \mathcal{P}_x| \leq \ell$, we consider $\Phi(x, y, h) \in \mathbb{R}^\ell$. Although we can consider $\boldsymbol{w} \in \mathbb{R}^\ell$, the vector $\boldsymbol{w}$ is sparse with at most $\mathfrak{s}$ non-zero entries. Thus, we take into account all possible subsets of $\mathfrak{s}$ features from $\ell$ possible features. From results in [2, 3, 8], we can conclude that there are at most $(nr)^{2(\mathfrak{s}-1)}$ linearly inducible orderings, for a fixed set of $\mathfrak{s}$ features. Therefore, there are at most $\binom{\ell}{\mathfrak{s}}(nr)^{2(\mathfrak{s}-1)} \leq \ell^{\mathfrak{s}}(nr)^{2\mathfrak{s}}$ collections $\Pi$.

Fix $\delta \in (0, 1)$. By Rademacher-based uniform convergence[3] and by a union bound over all $\ell^{\mathfrak{s}}(nr)^{2\mathfrak{s}}$ collections $\Pi$, with probability at least $1 - \delta/2$ over the choice of $n$ sets of random structured outputs, simultaneously for all parameters $\boldsymbol{w} \in \mathcal{W}$:

$$B(\boldsymbol{w}, S, \mathbb{T}(\boldsymbol{w})) \leq 2\,\mathfrak{R}_{\mathbb{T}(\Pi)}(\mathfrak{G}_{\mathcal{W},\Pi,S}) + 3\sqrt{\frac{\mathfrak{s}(\log \ell + 2\log(nr)) + \log(4/\delta)}{n}} \tag{18}$$

where $\mathfrak{R}_{\mathbb{T}(\Pi)}(\mathfrak{G}_{\mathcal{W},\Pi,S})$ is the *empirical* Rademacher complexity of the function class $\mathfrak{G}_{\mathcal{W},\Pi,S}$ with respect to the collection $\mathbb{T}(\Pi)$ of the $n$ sets $T(\pi(x), x)$ for all $(x, y) \in S$. Let $\sigma$ be an $n$-dimensional vector of independent Rademacher random variables indexed by $(x, y) \in S$, i.e., $\mathbb{P}[\sigma_{(x,y)} = +1] = \mathbb{P}[\sigma_{(x,y)} = -1] = 1/2$. The empirical Rademacher complexity is defined as:

$$\mathfrak{R}_{\mathbb{T}(\Pi)}(\mathfrak{G}_{\mathcal{W},\Pi,S}) \equiv \mathop{\mathbb{E}}_{\sigma} \left[ \sup_{g \in \mathfrak{G}_{\mathcal{W},\Pi,S}} \left( \frac{1}{n} \sum_{(x,y) \in S} \sigma_{(x,y)} g(x, y, T(\pi(x), x), \boldsymbol{w}) \right) \right]$$

$$= \mathop{\mathbb{E}}_{\sigma} \left[ \sup_{\boldsymbol{w} \in \mathcal{W}_{\Pi,S}} \left( \frac{1}{n} \sum_{(x,y) \in S} \sigma_{(x,y)} \max_{(\hat{y}, \hat{h}) \in T(\pi(x), x)} d(y, \hat{y}, \hat{h}) \, \mathbb{1}\left[ 1 - \widetilde{m}(x, y, \hat{y}, \hat{h}, \boldsymbol{w}) \geq 0 \right] \right) \right]$$

$$= \mathop{\mathbb{E}}_{\sigma} \left[ \sup_{\boldsymbol{w} \in \mathcal{W}_{\Pi,S}} \left( \frac{1}{n} \sum_{(x,y) \in S} \sigma_{(x,y)} \max_{(\hat{y}, \hat{h}) \in T(\pi(x), x)} d(y, \hat{y}, \hat{h}) \, \mathbb{1}\left[ 1 \geq \max_{h \in \widetilde{\mathcal{H}}_x} \Phi(x, y, h) \cdot \boldsymbol{w} - \Phi(x, \hat{y}, \hat{h}) \cdot \boldsymbol{w} \right] \right) \right]$$

$$= \mathop{\mathbb{E}}_{\sigma} \left[ \sup_{\boldsymbol{w} \in \mathbb{R}^\ell \backslash \{0\}} \left( \frac{1}{n} \sum_{i \in \{1 \ldots n\}} \sigma_i \max_{j \in \{1 \ldots n'\}} d_{ij} \, \mathbb{1}\left[ 1 \geq \max_{h \in \{1 \ldots |\widetilde{\mathcal{H}}_x|\}} z'_{ih} \cdot \boldsymbol{w} - z_{ij} \cdot \boldsymbol{w} \right] \right) \right]$$

$$\tag{19.a}$$

$$\leq \sum_{j\in\{1...n'\}} \mathop{\mathbb{E}}_{\sigma} \left[ \sup_{\boldsymbol{w}\in\mathbb{R}^{\ell}\setminus\{0\}} \left( \frac{1}{n} \sum_{i\in\{1...n\}} \sigma_i \, d_{ij} \, \mathbb{1}\left[1 \geq \max_{h\in\{1...|\widetilde{\mathcal{H}}_x|\}} z'_{ih}\cdot\boldsymbol{w} - z_{ij}\cdot\boldsymbol{w}\right] \right) \right]$$

$$\text{(19.b)}$$

$$\leq \sum_{j\in\{1...n'\}} \mathop{\mathbb{E}}_{\sigma} \left[ \sup_{\boldsymbol{w}\in\mathbb{R}^{\ell}\setminus\{0\}} \left( \frac{1}{n} \sum_{i\in\{1...n\}} \sigma_i \, \mathbb{1}\left[1 \geq \max_{h\in\{1...|\widetilde{\mathcal{H}}_x|\}} z'_{ih}\cdot\boldsymbol{w} - z_{ij}\cdot\boldsymbol{w}\right] \right) \right]$$

$$\text{(19.c)}$$

$$\leq \sum_{j\in\{1...n'\}} \mathop{\mathbb{E}}_{\sigma} \left[ \sup_{\tilde{\boldsymbol{w}}\in\mathbb{R}^{\ell(|\widetilde{\mathcal{H}}|+1)+1}\setminus\{0\}} \left( \frac{1}{n} \sum_{i\in\{1...n\}} \sigma_i \, \mathbb{1}\left[z_{ij}^{\widetilde{\mathcal{H}}}\cdot\tilde{\boldsymbol{w}} \geq 0\right] \right) \right] \qquad \text{(19.d)}$$

$$\leq 2n'\sqrt{\frac{(2\mathfrak{s}+1)\log\left(\ell(n\tilde{r}+1)+1\right)\log\left(n+1\right)}{n}} \qquad \text{(19.e)}$$

where in the step in eq.(19.a), the terms $\sigma_i, d_{ij}, z'_{ih}, z_{ij}$ correspond to $\sigma_{(x,y)}, d(y,\hat{y},\hat{h}), \Phi(x,y,h)$ and $\Phi(x,\hat{y},\hat{h})$ respectively. Thus, we assume that index $i$ corresponds to the training sample $(x,y)\in S$, and that index $j$ corresponds to the structured output and latent variable $(\hat{y},\hat{h})\in T(\pi(x),x)$. Note that since $\Phi(x,y,h)\in\mathbb{R}^{\ell}$, thus the step in eq.(19.a) considers $\boldsymbol{w}, z'_{ih}, z_{ij}\in\mathbb{R}^{\ell}\setminus\{0\}$ without loss of generality. The step in eq.(19.b) follows from the fact that for any two function classes $\mathfrak{G}$ and $\mathcal{H}$, we have that $\mathfrak{R}(\{\max(g,h)\mid g\in\mathfrak{G}\text{ and }h\in\mathcal{H}\})\leq\mathfrak{R}(\mathfrak{G})+\mathfrak{R}(\mathcal{H})$. The step in eq.(19.c) follows from the composition lemma and the fact that $d_{ij}\in[0,1]$ for all $i$ and $j$. The step in eq.(19.d) considers a larger function class, we consider $\tilde{\boldsymbol{w}}, z_{ij}^{\widetilde{\mathcal{H}}}\in\mathbb{R}^{\ell(|\widetilde{\mathcal{H}}|+1)+1}\setminus\{0\}$. More detailed, for a fixed $i,j$, and $\boldsymbol{w}\in\mathbb{R}^{\ell}$, we can construct the vectors $z_{ij}^{\widetilde{\mathcal{H}}} = (1, -z'_{i1}, \ldots, -z'_{i|\widetilde{\mathcal{H}}|}, z_{ij})$ and $\tilde{\boldsymbol{w}}^{(t)} = (1, \boldsymbol{w}^{(1)}, \ldots, \boldsymbol{w}^{(|\widetilde{\mathcal{H}}|)}, \boldsymbol{w})$, where $\boldsymbol{w}^{(l)}=\boldsymbol{w}$ if $l=t$, and $\boldsymbol{w}^{(l)}=\boldsymbol{0}$ otherwise. The step in eq.(19.e) follows from the Massart lemma, the Sauer-Shelah lemma and the VC-dimension of sparse linear classifiers. That is, for any function class $\mathfrak{G}$, we have that $\mathfrak{R}(\mathfrak{G})\leq\sqrt{\frac{2VC(\mathfrak{G})\log(n+1)}{n}}$ where $VC(\mathfrak{G})$ is the VC-dimension of $\mathfrak{G}$. Finally, note that $|\widetilde{\mathcal{H}}_x|\leq\tilde{r}, \forall(x,y)\in S$, and $|\widetilde{\mathcal{H}}|=|\cup_{(x,y)\in S}\widetilde{\mathcal{H}}_x|\leq n\tilde{r}$. Also, since $\boldsymbol{w}$ is $\mathfrak{s}$-sparse, we have that $\tilde{\boldsymbol{w}}$ is $(2\mathfrak{s}+1)$-sparse. Then, by Theorem 20 of [18], $VC(\mathfrak{G})\leq 2(2\mathfrak{s}+1)\log(\ell(|\widetilde{\mathcal{H}}|+1)+1)$ for the class $\mathfrak{G}$ of sparse linear classifiers on $\mathbb{R}^{\ell(|\widetilde{\mathcal{H}}|+1)+1}$, with $3\leq 2\mathfrak{s}+1\leq\frac{9}{20}\sqrt{\ell(|\widetilde{\mathcal{H}}|+1)+1}$.

By eq.(11), eq.(13.c), eq.(17.c), eq.(18) and eq.(19.e), we prove our claim. $\qquad\square$

### A.4 Proof of Claim i

*Proof.* For all $(x,y)\in S$ and $\boldsymbol{w}\in\mathcal{W}$, by definition of the total variation distance, we have for any event $\mathcal{A}(x,y,y',h',\boldsymbol{w})$:

$$\left| \mathop{\mathbb{P}}_{(y',h')\sim R(\boldsymbol{w},x)}[\mathcal{A}(x,y,y',h',\boldsymbol{w})] - \mathop{\mathbb{P}}_{(y',h')\sim R'(\boldsymbol{w},x)}[\mathcal{A}(x,y,y',h',\boldsymbol{w})] \right| \leq TV(R(\boldsymbol{w},x)\|R'(\boldsymbol{w},x))$$

Let the event $\mathcal{A}(x,y,y',h',\boldsymbol{w}): d(y,y',h')=1$ and $1-m(x,y,y',h',\boldsymbol{w})\geq 0$. Since $R(\boldsymbol{w},x)$ fulfills Assumption A with value $\beta_1$ and since $TV(R(\boldsymbol{w},x)\|R'(\boldsymbol{w},x))\leq\beta_2$, we have that for all $(x,y)\in S$ and $\boldsymbol{w}\in\mathcal{W}$:

$$\mathop{\mathbb{P}}_{(y',h')\sim R'(\boldsymbol{w},x)}[\mathcal{A}(x,y,y',h',\boldsymbol{w})] \geq \mathop{\mathbb{P}}_{(y',h')\sim R(\boldsymbol{w},x)}[\mathcal{A}(x,y,y',h',\boldsymbol{w})] - TV(R(\boldsymbol{w},x)\|R'(\boldsymbol{w},x))$$

$$\geq 1 - \beta_1 - \beta_2$$

which proves our claim. $\qquad\square$

### A.5 Proof of Claim ii

*Proof.* Since $\mathcal{Y}_x$ is the set of all permutations of $v$ elements, then $|\mathcal{Y}_x|=v!$. In addition, since $d(y,y',h)=\frac{1}{v}\sum_{i=1}^{v}\mathbb{1}[y_i\neq y'_i]$ and since $R(x)$ is a uniform proposal distribution with support on

$\mathcal{Y}_x \times \mathcal{H}_x$, we have:

$$\mathbb{P}_{(y',h')\sim R(x)}[d(y,y',h')=1] = \mathbb{P}_{y'}[d(y,y')=1]$$

$$= \frac{F(v)}{v!} \tag{20.a}$$

$$\geq 1 - 2/3.$$

For a fixed $y$, the function $F(v)$ in step eq.(20.a) represents the number of permutations $y' \in \mathcal{Y}_x$ such that $d(y, y', h) = 1$. Moreover, $F(v)$ can be computed through the following recursion: $F(v) = (v-1)! \times (1 + \sum_{i=1}^{v-2} \frac{F(i)}{i!})$. The probability is then $F(v)/v!$, it can be seen that this probability converges as $v \to \infty$ through the following: $\lim_{v\to\infty} \frac{F(v+1)}{(v+1)!} - \frac{F(v)}{v!} = 0$. The probability converges to 0.3679 approximately, while achieving a minimum value of $1/3$ at $v = 3$. Hence $\beta = 2/3$. $\quad\square$

## A.6 Proof of Claim iii

*Proof.* Let $\Delta \equiv \Phi(x, y, h^*) - \Phi(x, y', h')$. Let $p \in \mathcal{P}_x$ be a superindex denoting the partitions, i.e., for all $p \in \mathcal{P}_x$, let $\Delta^p \equiv \Phi(x, y, h^*) - \Phi(x, y', h')$ for some $(y', h') \in \Upsilon_x^p$. By assumption, since $(y', h') \in \Upsilon_x^p$ then $|\Delta_p^p| \leq b$ and $(\forall q \neq p)\, \Delta_q^p = 0$. Therefore:

$$\left\| \mathbb{E}_{(y',h')\sim R(x)}[\Delta] \right\|_2 = \sqrt{\sum_{q\in\mathcal{P}_x} \mathbb{E}_{(y',h')\sim R(x)}[\Delta_q]^2}$$

$$\leq \sqrt{\sum_{q\in\mathcal{P}_x} \mathbb{E}_{(y',h')\sim R(x)}[|\Delta_q|]^2}$$

$$= \sqrt{\sum_{q\in\mathcal{P}_x} \left( \sum_{p\in\mathcal{P}_x} \mathbb{P}_{(y',h')\sim R(x)}[(y',h')\in\Upsilon_x^p]\, |\Delta_q^p| \right)^2}$$

$$= \sqrt{\sum_{q\in\mathcal{P}_x} \left( \mathbb{P}_{(y',h')\sim R(x)}[(y',h')\in\Upsilon_x^q]\, |\Delta_q^q| \right)^2}$$

$$\leq \sqrt{|\mathcal{P}_x| \left( \frac{b}{|\mathcal{P}_x|} \right)^2}$$

$$= b/\sqrt{|\mathcal{P}_x|}$$

where we used the fact that for a uniform proposal distribution $R(x)$, we have $\mathbb{P}_{(y',h')\sim R(\boldsymbol{w},x)}[(y',h')\in\Upsilon_x^q] = 1/|\mathcal{P}_x|$. Finally, since we assume that $n \leq |\mathcal{P}_x|/(4b^2)$, we have $b/\sqrt{|\mathcal{P}_x|} \leq 1/(2\sqrt{n})$ and we prove our claim. $\quad\square$

## A.7 Proof of Claim iv

*Proof.* Let $\Delta \equiv \Phi(x, y, h^*) - \Phi(x, y', h')$. By assumption $|\Delta_p| \leq b/|\mathcal{P}_x|$ for all $p \in \mathcal{P}_x$. Therefore:

$$\left\| \mathbb{E}_{(y',h')\sim R(\boldsymbol{w},x)}[\Delta] \right\|_2 = \sqrt{\sum_{p\in\mathcal{P}_x} \mathbb{E}_{(y',h')\sim R(\boldsymbol{w},x)}[\Delta_p]^2}$$

$$\leq \sqrt{\sum_{p\in\mathcal{P}_x} \mathbb{E}_{(y',h')\sim R(\boldsymbol{w},x)}[|\Delta_p|]^2}$$

$$\leq \sqrt{|\mathcal{P}_x| \left( \frac{b}{|\mathcal{P}_x|} \right)^2}$$

$$= b/\sqrt{|\mathcal{P}_x|}$$

Finally, since we assume that $n \leq |\mathcal{P}_x|/(4b^2)$, we have $b/\sqrt{|\mathcal{P}_x|} \leq 1/(2\sqrt{n})$ and we prove our claim. $\quad\square$

### A.8 Proof of Claim v

*Proof.* Algorithm 1 depends solely on the linear ordering induced by the parameter $\boldsymbol{w}$ and the mapping $\Phi(x, \cdot)$. That is, at any point in time, Algorithm 1 executes comparisons of the form $\Phi(x, y, h) \cdot \boldsymbol{w} > \Phi(x, \hat{y}, \hat{h}) \cdot \boldsymbol{w}$ for any two pair of structured outputs and latent variables $(y, h)$ and $(\hat{y}, \hat{h})$. $\square$

## Appendix B    Discussion, Further Examples and Details of Experiments

### B.1    Discussion

In this section, we discuss in more detail the inference problem. We also briefly discuss the non-convexity of the formulation in eq.(6).

**Inference on Test Data.**    The upper bound in Theorem 3 holds simultaneously for all parameters $\boldsymbol{w} \in \mathcal{W}$. Therefore, our result implies that after learning the optimal parameter $\hat{\boldsymbol{w}} \in \mathcal{W}$ in eq.(6) from *training* data, we can bound the decoder distortion when performing *exact* inference on *test* data. More formally, Theorem 3 can be additionally invoked for a *test* set $S'$, also with probability at least $1 - \delta$. Thus, under the same setting as of Theorem 3, the Gibbs decoder distortion is upper-bounded with probability at least $1 - 2\delta$ over the choice of $S$ and $S'$. In this paper, we focus on learning the parameter of structured prediction models. We leave the analysis of approximate inference on test data for future work.

**A Non-Convex Formulation.**    As mentioned in Section 2, all formulations with latent variables (eq.(4),eq.(5), and eq.(6)) are non-convex objectives. The motivation to use the margin re-scaling approach in the work of Yu and Joachims [30] is that the non-convex objective leads to a difference of two convex functions, which allows the use of CCCP [31]. In the case of models without latent variables, Sarawagi and Gupta [24] propose a method to reduce the problem of slack re-scaling to a series of modified margin re-scaling problems. However, there are two main caveats in their approach. First, the optimization is only heuristic, that is, it is not guaranteed to solve the slack rescaling objective exactly. Second, their method is specific to the cutting plane training algorithm and does not easily extend to stochastic algorithms. Choi et al. [4] propose efficient methods for finding the most-violating-label in a slack re-scaling formulation, given an oracle that returns the most-violating-label in a (slightly modified) margin re-scaling formulation. However, in the case of latent models, it is still unclear if this sort of reductions are possible for the slack re-scaling approach because of the maximization in the margin with respect to the latent space.

We also note that one way to make the objective in eq.(5) convex is to replace the maximization in the margin by the latent variable $\hat{h}$. However, this not only results in a looser upper bound of the Gibbs decoder distortion but also under performs with respect to the methods mentioned in this paper.

**Randomizing the Latent Space.**    We note that in the definition of the margin, there is a maximization over the latent space $\mathcal{H}$. In this paper, we sample structured outputs and latent variables from some proposal distribution and these samples are used in the outer maximization in eq.(6). While sampling latent variables from some proposal distribution in the maximization of the margin might be computationally appealing, the main issue is that this will lead to a looser upper bound of the Gibbs decoder distortion.

### B.2    Further examples for Assumption A

For completeness, we present the examples provided in [11] since we make use of the suggested $\beta$ values in our synthetic experiments. Although their proofs are given without using latent variables, it is straightforward to extend their claims by marginalizing on $h$.

**Any type of structured output for binary distortion functions.**    Let $\mathcal{Y}_x \times \mathcal{H}_x$ be an arbitrary countable set of feasible decodings of $x$, such that $|\mathcal{Y}_x| \geq 2$ for all $(x, y) \in S$. Let $d(y, y', h) = \mathbb{1}[y \neq y']$. The uniform proposal distribution $R(\boldsymbol{w}, x) = R(x)$ with support on $\mathcal{Y}_x \times \mathcal{H}_x$ fulfills Assumption A with $\beta = 1/2$.

**Directed spanning trees for a distortion function that returns the number of different edges.**
Let $\mathcal{Y}_x$ be the set of directed spanning trees of $v$ nodes. Let $A(y)$ be the adjacency matrix of $y \in \mathcal{Y}_x$. Let $d(y, y', h) = \frac{1}{2(v-1)} \sum_{ij} |A(y)_{ij} - A(y')_{ij}|$. The uniform proposal distribution $R(\boldsymbol{w}, x) = R(x)$ with support on $\mathcal{Y}_x \times \mathcal{H}_x$ fulfills Assumption A with $\beta = \frac{v-2}{v-1}$.

**Directed acyclic graphs for a distortion function that returns the number of different edges.**
Let $\mathcal{Y}_x$ be the set of directed acyclic graphs of $v$ nodes and $b$ parents per node, such that $2 \leq b \leq v - 2$. Let $A(y)$ be the adjacency matrix of $y \in \mathcal{Y}_x$. Let $d(y, y', h) = \frac{1}{b(2v-b-1)} \sum_{ij} |A(y)_{ij} - A(y')_{ij}|$. The uniform proposal distribution $R(\boldsymbol{w}, x) = R(x)$ with support on $\mathcal{Y}_x \times \mathcal{H}_x$ fulfills Assumption A with $\beta = \frac{b^2 + 2b + 2}{b^2 + 3b + 2}$.

**Cardinality-constrained sets for a distortion function that returns the number of different elements.** Let $\mathcal{Y}_x$ be the set of sets of $b$ elements chosen from $v$ possible elements, such that $b \leq v/2$. Let $d(y, y', h) = \frac{1}{2b}(|y - y'| + |y' - y|)$. The uniform proposal distribution $R(\boldsymbol{w}, x) = R(x)$ with support on $\mathcal{Y}_x \times \mathcal{H}_x$ fulfills Assumption A with $\beta = 1/2$.

### B.3 Additional Details of Experiments

**Synthetic Experiments.** We replaced the discontinuous 0/1 loss $\mathbb{1}[z \geq 0]$ with the convex hinge loss $\max(0, 1 + z)$, as it is customary. Note however, that even by using the hinge loss, the objective functions in eq.(4), eq.(5) and in eq.(6) are still non-convex with respect to $\boldsymbol{w}$. This is due to the maximization over the latent space in the definition of the margin. We used $\lambda = 1/n$ as suggested by Theorems 1 and 3, and we performed 30 iterations of the subgradient descent method with a decaying step size $1/\sqrt{t}$ for iteration $t$. For sampling random structured outputs and latent variables in eq.(6), we implemented Algorithm 1 for directed spanning trees, directed acyclic graphs and cardinality-constrained sets. We performed the local changes in Algorithm 1 as follows. Given a pair $(\hat{y}, \hat{h})$, making a local change to $(\hat{y}, \hat{h})$ consists on iterating through all pairs $(y', h')$ where $\hat{y}$ and $y'$ differ only in one edge/element, and where the single entries in $\hat{h}$ and $h'$ are contiguous. Finally, we used $\beta = 0.67$ for directed spanning trees, $\beta = 0.84$ for directed acyclic graphs, and $\beta = 0.5$ for cardinality-constrained sets, as prescribed by the examples given in Section B.2.

**Image Matching.** Ground truth is provided in the Buffy Stickmen dataset for measuring performance on a test set. The authors in [9, 27] did not use latent variables, and considered the mapping $\Phi(x, y) = \frac{1}{18} \sum_{i=1}^{18} (\psi(I, i) - \psi(I', y_i))^2$, where $\psi(I, k) \in \mathbb{R}^{128}$ are the SIFT descriptors at scale 5 evaluated at keypoint $k$. We properly centered the coordinates independently on each frame to avoid modeling translations in $h$. We use the mapping $\Phi(x, y, h) = (\Phi(x, y), \frac{1}{18} \sum_{i=1}^{18} \|c(I, i) \times h - c(I', y_i)\|_2^2)$, where $c(I, k) \in \mathbb{R}^2$ are the coordinates of keypoint $k$. Intuitively, we are adding one extra feature that summarizes the change in rotation and scaling of the keypoints, i.e., $\Phi(x, y, h) \in \mathbb{R}^{129}$.

The learning is performed using the random formulation as in eq.(6), and using local changes as in Algorithm 1 for sampling from the proposal distribution. As in the synthetic experiments, we also replaced the discontinuous 0/1 loss $\mathbb{1}[z \geq 0]$ with the convex hinge loss $\max(0, 1 + z)$, and followed the local changes in Algorithm 1 for sampling from the proposal distribution. The neighborhoods of the structures and latent variables were defined as follow: for a given permutation $y$, we considered $y'$ to be its neighbor, and vice versa, if they have only two mismatched entries. Similarly, for a given $h$, we considered $h'$ to be its neighbor, and vice versa, if they have only one different entry.

## Footnotes

[2] The same result can be obtained for any subset of $S$ for which the "separability" condition holds. Therefore, our analysis with the "non-separability" condition can be seen as a worst case scenario.

[3] Note that for the analysis of $B(\boldsymbol{w}, S, \mathbb{T}(\boldsymbol{w}))$, the training set $S$ is fixed and randomness stems from the collection $\mathbb{T}(\boldsymbol{w})$. Also, note that for applying McDiarmid's inequality, independence of each set $T(\boldsymbol{w}, x)$ for all $(x, y) \in S$ is a sufficient condition, and identically distributed sets $T(\boldsymbol{w}, x)$ are not necessary.