[Reviews · NeurIPS 2018]

Reviewer 1



This paper extends recent ideas in training structured predictors using randomized algorithms to the latent variable setting. Traditional large-margin training for structured prediction defines the loss function in terms of constraints or an optimization over the entire structured output space, which in general makes inference and learning NP-hard. Recent work has considered defining the loss over only samples in the output space from a proposal distribution. In this case, the approximate loss function bounds a robust learning objective over the entire output space up to known approximation error. This paper considers extending this technique and analysis to the setting where there are latent variables. The key technical contribution is an analysis that is tighter than a naive application of Rademacher complexity analysis, leading to a sample complexity that is only logarithmic in the size of the latent space. Experiments show that the proposed method is as accurate but >10x faster than traditional large-margin learning techniques on synthetic data and an image alignment problem. The paper is clear, and makes an interesting contribution. Update: Thank you to the authors for their response. The other reviewers raised good points about clarifications that would enhance the paper. I encourage the authors to take those suggestions.

Reviewer 2



Overview: This paper presents an approach to train latent structural SVMs. The objective function is based on a slack rescaling loss which uses a latent margin form (eq 5). The maximization over all possible outputs in the loss is replaced with a maximization over a (small) random sample of structured outputs from a proposal distribution (that satisfies a few assumptions). A PAC-Bayes generalization bound is derived, with robustness to Gaussian perturbations. Assuming that the maximization over the latent space is tractable, this approach yields an efficient training algorithm with generalization guarantees. The paper seems to be technically sound and the motivation is clear, although the proofs are quite involved and I have not verified all of them carefully. The main limitation in my opinion is incremental contribution over the work of Honorio & Jaakkola [11], possibly better as an extended journal version of [11] than a standalone conference paper? Detailed comments: * This work follows [11] very closely. The structure and theorems are analogous to [11]. Some sentences are even copied as is -- for example, lines 71, 86-87, 125,... * Line 95: an example of “parts” can be very helpful for clarity. * For footnote 1, I suggest to move it to the main text and add an example in section 5. This is essentially an additional assumption -- necessary for obtaining a tractable algorithm. What if the maximum can be approximated efficiently? Do some guarantees still hold? * In the image-matching experiment, is it possible to add results for an LSSVM or other baseline besides [9]? * Line 206: I suggest to provide high-level intuition of the idea behind the tighter bound, the current wording is rather mysterious. * For training, is the upper bound in Thm 2 minimized? Adding the details of the training algorithm (perhaps in the appendix for lack of space) would improve readability.

Reviewer 3



UPDATE: Upon reading the author response, I have decided to leave my review unchanged. The topic of this work is of interest to the community, and it can make a decent publication. I am still concerned about novelty. While the author response argued that similar increments have been done in past works, I feel that there are some significant differences between the current case and the mentioned works. Accepting the paper is still a reasonable decision in my humble opinion. Original Review: The paper develops PAC-Bayesian bounds for margin-based structured prediction, when hidden variables are included in the problem. It mainly relies on the analysis of [1], adapted to the formulation of structured SVM with latent variables given in [2]. The point I found most interesting is the remark about attempts to convexify structured losses studied in the paper. The necessity for a non-convex formulation stems directly from the PAC-Bayesian bound, which is a nice and novel insight. My main concern regarding this paper is whether it is sufficiently novel. The majority of technical work is an almost direct adaptation of [1], and the paper does not seem to offer a significantly novel approach for the analysis of structured prediction problems. With that said, the problem of latent variables in structured prediction is interesting. This is enough to shift my overall score to 6. The paper could also gain from a more thorough experimental study. In all the proposed experiments, the latent space is very small and marginalization or enumeration of the hidden states is easy. I think that a problem with a large latent space, where the randomization of this space gives a significant advantage, could strengthen the point of the paper. This would also distinguish the algorithmic contribution of this paper from the algorithms analyzed in [1] and demonstrated in papers on natural language processing. Overall, I think that while this paper makes nice contributions, it is not clear if they amount to a significantly novel submission. [1] Honorio, J. and Jaakkola, T. Structured prediction: from Gaussian perturbations to linear-time principled algorithms. UAI 2016 [2] Yu, C. and Joachims, T. Learning structural SVMs with latent variables. ICML 2009